# Whatever Remains Must Be True: Filtering Drives Reasoning in LLMs, Shaping Diversity

**Germán Kruszewski**[1]\*, **Pierre Erbacher**[1]\*, **Jos Rozen**[1] **& Marc Dymetman**[2]

[1]Naver Labs Europe    [2]Independent Researcher

{german.kruszewski,pierre.erbacher,jos.rozen}@naverlabs.com
marc.dymetman@gmail.com

## Abstract

Reinforcement Learning (RL) has become the *de facto* standard for tuning LLMs to solve tasks involving reasoning. However, growing evidence shows that models trained in such way often suffer from a significant loss in diversity. We argue that this arises because RL implicitly optimizes the "mode-seeking" or "zero-forcing" *Reverse KL* to a target distribution causing the model to concentrate mass on certain high-probability regions of the target while neglecting others. In this work, we instead begin from an explicit target distribution, obtained by filtering out incorrect answers while preserving the relative probabilities of correct ones. Starting from a pre-trained LLM, we approximate this target distribution using the $\alpha$-divergence family, which unifies prior approaches and enables direct control of the precision–diversity trade-off by interpolating between mode-seeking and mass-covering divergences. On a LEAN theorem-proving benchmark, our method achieves state-of-the-art performance along the coverage–precision Pareto frontier, outperforming all prior methods on the coverage axis.

## 1 Introduction

> "How often have I said to you that when you have eliminated the impossible, whatever remains, *however improbable*, must be the truth?"
>
> — **Arthur Conan Doyle**, *The Sign of Four*

Large Language Models (LLMs) have made striking progress on reasoning tasks. A leading approach is Reinforcement Learning from Verifiable Rewards (RLVR) (Lambert et al., 2025; DeepSeek-AI et al., 2025), where policy-gradient methods such as PPO (Schulman et al., 2017) or GRPO (Shao et al., 2024) optimize against a reward that combines a binary verifier of correctness with a KL penalty to keep the tuned model close to its base distribution.

While RLVR has been credited with enabling exploration of new solutions (DeepSeek-AI et al., 2025), recent studies challenge this view. In particular, they find that base models already contain these solutions given a sufficient sampling budget, and that tuned models often exploit additional samples less effectively due to reduced output diversity (Yue et al., 2025b; He et al., 2025; Dang et al., 2025; Wu et al., 2025). Earlier, with RLHF (Christiano et al., 2017; Ziegler et al., 2020), similar diversity reductions were observed, sometimes described as "mode collapse" (Kirk et al., 2024; O'Mahony et al., 2024).

We argue that this loss of diversity stems from the *implicit objective* of RL-based training: optimizing the *Reverse KL* divergence to a target distribution that favors correct answers (Korbak et al., 2022b). Reverse KL is "mode-seeking" or "zero-forcing," emphasizing precision on a subset of solutions while ignoring others (Bishop, 2006; Huszár, 2015; Li & Farnia, 2023). This explains why RLVR models become accurate but less diverse.

---

\*Equal contribution

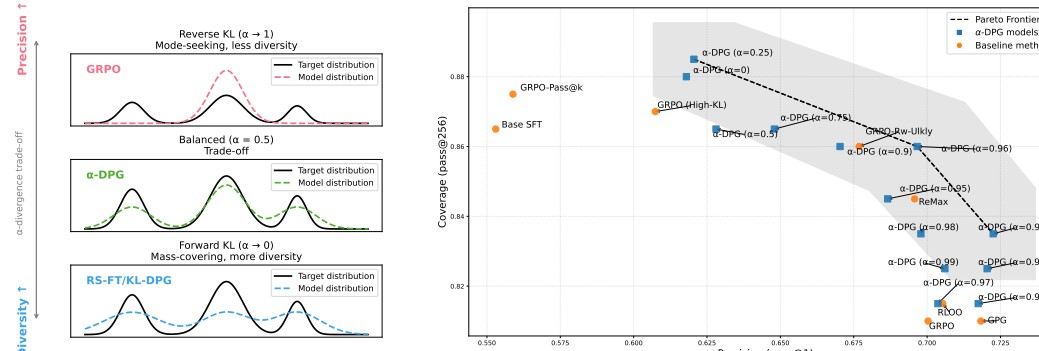

Figure 1: **Left:** Illustration of our method. RLVR policy-gradient approaches (e.g., GRPO, PPO) concentrate probability on a narrow region of the target distribution, while methods like KL-DPG increase diversity but allocate mass to low-quality regions. $\alpha$-DPG balances these extremes. **Right:** Model precision (pass@1) and coverage (pass@256). The Pareto frontier is the upper-right boundary of the convex hull; the shaded band shows one bootstrap standard deviation. $\alpha$-DPG and baseline models lie along the frontier, with $\alpha$-DPG achieving the highest coverage and precision for low and high values of $\alpha$, respectively.

To address this, we explicitly define the desired target distribution: one that *always* outputs correct solutions while remaining as close as possible to the base model, thus preserving any solution included therein (Khalifa et al., 2021; Kim et al., 2025). Direct sampling is infeasible, but we can approximate it with an autoregressive policy using Distributional Policy Gradient Algorithms (DPG Parshakova et al., 2019; Khalifa et al., 2021; Go et al., 2023). Concretely, we apply $f$-DPG (Go et al., 2023), which minimizes an $f$-divergence (Polyanskiy & Wu, 2025) to this target. Different divergences trade off precision (probability of sampling a correct solution) and coverage (probability of sampling at least one correct solution given a sufficiently large sampling budget): Reverse KL emphasizes the former, Forward KL the latter (Bishop, 2006). To interpolate between them, we introduce $\alpha$-DPG, based on $\alpha$-divergences (Rényi, 1961; Cressie & Read, 1984; Amari, 1985). Notably, this method unifies RLVR (Reverse KL) and both KL-DPG (Parshakova et al., 2019; Khalifa et al., 2021; Go et al., 2023) and Rejection Sampling Fine-Tuning (Forward KL) (Zelikman et al., 2022; Yuan et al., 2023) under a single umbrella. We call this approach Distributional Matching with Verifiable Rewards (DMVR), positioning it under the general framework of Distributional Matching (Khalifa et al., 2021; Korbak et al., 2022a;b).

We evaluate $\alpha$-DPG in LEAN, a proof assistant that automatically verifies formal mathematical proofs. In this setting, success requires not only correct candidates but also diversity across proof attempts, since harder theorems may be solvable only through rare derivations. We find that $\alpha$-DPG achieves state-of-the-art performance, producing models that lie on the Pareto frontier between precision (pass@1) and coverage (pass@256), and that surpass prior methods in coverage[1].

**Contributions.**

- We introduce the DMVR framework, which trains models by approximating an explicitly defined verifier-based target distribution.

- We clarify how the implicit dynamics of RL-based methods lead to reduced diversity.

- We highlight the role of the divergence family in trading off precision and diversity, and propose $\alpha$-DPG to smoothly interpolate between Forward and Reverse KL.

- We show on the LEAN benchmark that $\alpha$-DPG achieves results along a Pareto coverage-precision frontier, with the $\alpha$ parameter allowing to trade-off between precision and coverage. Moreover, $\alpha$-DPG achieves the best coverage results among all considered methods when using low values of $\alpha$.

---

[1]We make the code available at https://github.com/naver/alpha-dpg.

## 2 BACKGROUND

**Reinforcement Learning with Verifiable Rewards (RLVR)** Let $\pi_\theta(\cdot|x) : \mathcal{Y} \to \mathbb{R}$ be an LLM with parameters $\theta$ defining a probability distribution over sequences $y \in \mathcal{Y}$ conditioned on a prompt $x$. A verifier $v(y, x) \in \{0, 1\}$ is a binary function that discriminates between correct and incorrect responses to $x$. Given a dataset $\mathcal{D}$ of prompts (and, optionally, ground-truth answers for the verifier), RLVR uses a standard reinforcement learning objective derived from previous work on Reinforcement Learning from Human Feedback (RLHF) (Christiano et al., 2017; Ziegler et al., 2020):

$$\mathcal{J}_{\text{RLVR}} = \arg\max_\theta \mathbb{E}_{x\sim\mathcal{D}, y\sim\pi_\theta(\cdot|x)} R_\theta(y, x), \tag{1}$$

where the "pseudo-reward" (Korbak et al., 2022b) $R_\theta$ is defined as follows:

$$R_\theta(y, x) = v(y, x) - \beta \log \frac{\pi_\theta(y|x)}{\pi_{\text{base}}(y|x)}. \tag{2}$$

Here, $\pi_{\text{base}}$ is the base LLM and $\beta$ is a tunable parameter that controls the trade-off between maximizing the reward and minimizing divergence from the original model. Some authors fall back to setting $\beta = 0$, just optimizing the expected verifier reward (Liu et al., 2025; Mistral-AI et al., 2025; Yu et al., 2025).

**Policy Gradient (PG) Algorithms** We can maximize Eq. 2 following multiple algorithms. One of the simplest ones is KL-Control (Todorov, 2006; Kappen et al., 2012), with the following gradient

$$\nabla_\theta \mathcal{J}_{\text{KL-Control}}(\theta) = \mathbb{E}_{x\sim\mathcal{D}, y\sim\pi_\theta} \hat{A}(y, x) \nabla_\theta \log \pi_\theta(y|x), \tag{3}$$

where $\hat{A}(y, x) = R_\theta(y, x) - B(x)$ is the advantage function and $B(x)$ is a baseline that doesn't depend on $y$ to keep the objective unbiased used for variance reduction (Sutton & Barto, 2020, Chapter 2, Section 7). Some options for an unbiased baseline could include a constant, a critic (Sutton et al., 1999), or a leave-one-out average in a batch of rewards (RLOO; Kool, 2019; Ahmadian et al., 2024)[2]. When $\beta = 0$, Eq. 3 reduces to the original policy gradient algorithm REINFORCE (Williams, 1992). Another popular choice is PPO (Schulman et al., 2017), which optimizes the following clipped surrogate objective:

$$\mathcal{J}^{\text{PPO}}(\theta) = \mathbb{E}_{x\sim\mathcal{D}, y\sim\pi_{\theta_{old}}} \left[ \sum_{t=1}^{|y|} \min\left( \rho(y_t|x, y_{<t})\hat{A}(y, x), \text{clip}\left(\rho(y_t|x, y_{<t}), 1 - \epsilon, 1 + \epsilon\right)\hat{A}(y, x)\right)\right], \tag{4}$$

where $\rho(y_t|x, y_{<t}) = \pi_\theta(y_t|x, y_{<t})/\pi_{\theta_{old}}(y_t|x, y_{<t})$ is the ratio between the current and the previous policy token probabilities and $\epsilon$ is a small hyperparameter. PPO discourages large policy updates that would move $\rho$ outside the interval $[1-\epsilon, 1+\epsilon]$, which improves training stability by constraining the new policy to be close to the old one. GRPO (DeepSeek-AI et al., 2025) and other recent methods use the same clipping strategy, but differ in the form of the baseline. While PPO is commonly understood to use a critic model as a baseline, GRPO uses the group-level reward average.

**Distribution Matching (DM)** DM (Khalifa et al., 2021; Korbak et al., 2022a;b; Go et al., 2023; Kim et al., 2025) is a family of techniques for aligning LLMs to meet certain constraints. For instance, let's suppose that we want to constrain the LM so that all sequences $y$ meet $r(x, y) = 1$ for some binary filter $r(x, y) \in \{0, 1\}$[3]. The method expresses this as a target distribution we want to approximate, which in this case is given by

$$p_x(y) \propto \pi_{\text{base}}(y|x)\, r(y, x). \tag{5}$$

This distribution is the only distribution $p'$ that fulfills the following two desirable conditions: (i) it satisfies the constraint $r(y, x) = 1$, $\forall y \in \text{Supp}(p'(\cdot|x))$, and (ii) it is the closest to $\pi_{\text{base}}(\cdot|x)$ in terms of $D_{\text{KL}}(p'(\cdot|x)||\pi_{\text{base}}(\cdot|x))$. The second condition guarantees the preservation of all the diversity contained in the original model. In information geometric terms, $p_x$ is the I-projection of $\pi_{\text{base}}(\cdot|x)$ into the manifold of all distributions that satisfy the constraint given by $r$ (Csiszár & Shields, 2004).

---

[2]Note that the average of all samples, as is used in GRPO, is biased instead.

[3]If the reader notices a strong similarity between a constraint $r(x, y)$ and a verifier $v(x, y)$, this is no coincidence: It is exactly this connection that we will exploit in this paper.

**Distributional Policy Gradient (DPG) Algorithms** Given a target distribution, we can optimize an autoregressive policy $\pi_\theta$ to approximate it. Khalifa et al. (2021) used for this the DPG algorithm (Parshakova et al., 2019; Korbak et al., 2022a;b), later denoted KL-DPG (Go et al., 2023), which minimizes the Forward KL $D_{\text{KL}}(p_x||\pi_\theta)$ to the target distribution $p_x$:

$$\nabla_\theta \mathcal{L}^{\text{DPG}}(\theta) = \nabla_\theta \mathbb{E}_{x \sim \mathcal{D}} D_{\text{KL}}(p_x||\pi_\theta) = -\mathbb{E}_{x \sim \mathcal{D}, y \sim \pi_\theta(\cdot|x)} \left( \frac{p_x(y)}{\pi_\theta(y|x)} - 1 \right) \nabla_\theta \log \pi_\theta(y|x). \quad (6)$$

The negative one term acts as a constant baseline. Computing $p_x(y|x)$ requires estimating a normalization constant denoted as the partition function, $Z_x$, which can be estimated by importance sampling (Owen, 2013). Note that in the case of a binary constraint, $Z_x$ is the acceptance rate of the constraint when sampling from the base model (Kim et al., 2025): $Z_x = \sum_{y \in \mathcal{Y}} a(y|x) r(y, x) = \mathbb{E}_{y \sim a(\cdot|x)} r(y, x) = \mathbb{P}_{y \sim a(\cdot|x)}[r(y, x) = 1]$. (See App. E for additional details.)

Later, Go et al. (2023) introduced $f$-DPG, effectively generalizing this technique to any $f$-divergence by minimizing $D_f(\pi_\theta, p_x)$:

$$\nabla_\theta \mathcal{L}^{f\text{-DPG}}(\theta) = \nabla_\theta \mathbb{E}_{x \sim \mathcal{D}} \left[ D_f(\pi_\theta(\cdot|x)||p_x) \right] = \mathbb{E}_{x \sim \mathcal{D}, y \sim \pi_\theta(\cdot|x)} \left[ -\hat{A}^f(y, x) \nabla_\theta \log \pi_\theta(y|x) \right], \quad (7)$$

where $\hat{A}^f(y, x) = R_\theta^f(y, x) - B(x)$, $R_\theta^f(y, x) \doteq -f'\left(\frac{\pi_\theta(y|x)}{p_x(y)}\right)$ is a "pseudo-reward", $B(x)$ is a context-dependent baseline, and $f$ is a convex function that parametrizes the $f$-divergence where $f : (0, \infty) \to \mathbb{R}$ s.t. $f(1) = 0$. If $p_x(y) = 0$, then, $R_\theta^f(y, x) = -f'(\infty)$, where $f'(\infty) \doteq \lim_{t \to 0} t f(\frac{1}{t})$.

## 3 DISTRIBUTIONAL MATCHING WITH VERIFIABLE REWARDS (DMVR)

Here, we adopt the DM framework, and propose that the ideal target distribution for tuning a language model $\pi_{\text{base}}(\cdot|x)$ to solve problems $x \sim \mathcal{D}$ by means of a verifier $v(y, x) \in \{0, 1\}$ is simply the result of applying the verifier as a binary constraint, i.e. $r(y, x) = v(y, x)$, as follows:

$$p_x(y) \propto \pi_{\text{base}}(y|x) v(y, x) \; \forall x \in \mathcal{D}. \quad (8)$$

This distribution *filters out all incorrect responses, leaving out only correct ones with the same relative probabilities as the reference LLM*. This is the single distribution that (i) always answers correctly to $x$, and (ii) it is closest to the base model $\pi_{\text{base}}$ as measured by $D_{\text{KL}}(\cdot||\pi_{\text{base}})$ (Khalifa et al., 2021). In the following discussion, we argue that (1) the distribution approximated by RLVR is closely related, becoming equivalent to $p_x$ in the limit $\beta \to 0$, (2) for any *fixed* $\beta > 0$, RLVR optimizes the Reverse KL to a target distribution, which has a mode-seeking behavior, incentivizing the policy to put high probability mass in small regions of high reward at the cost of diversity, and (3) we propose an alternative based on $f$-DPG parametrized with $\alpha$-divergences to trade-off precision and diversity. Of these, points (1) and (3) are original to our work, whereas point (2) is reproduced from Korbak et al. (2022b). Moreover, the target distribution and the $f$-DPG technique to approximate it were defined in prior art (Khalifa et al., 2021; Go et al., 2023).

### 3.1 FROM RLVR TO DMVR

Consider the following lemma, reproducing the argument from Korbak et al. (2022b):

**Lemma 1.** *Define*

$$p_{x,\beta}(y) = \frac{1}{Z_x(\beta)} \pi_{base}(y \mid x) \exp\big(v(y, x)/\beta\big), \qquad Z_x(\beta) = \sum_y \pi_{base}(y \mid x) \exp\big(v(y, x)/\beta\big).$$

*Then,*

$$\nabla_\theta \mathbb{E}_x\big[\text{KL}(\pi_\theta||p_{x,\beta})\big] = -\mathbb{E}_{x \sim \mathcal{D}, y \sim \pi_\theta} \left[ \frac{1}{\beta} v(y, x) - \log \frac{\pi_\theta(y \mid x)}{\pi_{base}(y \mid x)} \right] \nabla_\theta \log \pi_\theta(y|x) \quad (9)$$

$$= -\frac{1}{\beta} \mathbb{E}_{x \sim \mathcal{D}, y \sim \pi_\theta} \left[ \underbrace{v(y, x) - \beta \log \frac{\pi_\theta(y \mid x)}{\pi_{base}(y \mid x)}}_{\text{RLVR pseudo-reward}} \right] \nabla_\theta \log \pi_\theta(y|x). \quad (10)$$

*(Proof in App. B).*

From Eq. 10, we can see that the gradient of the KL-Control's objective (right-hand-side, optimizing the expected RLVR pseudo-reward) is proportional to the gradient of the Reverse KL of $\pi_\theta$ to $p_{x,\beta}$ (left-hand-side), up to a negative constant that flips the direction of optimization. Therefore, maximizing the regularized reward implies minimizing the divergence, and vice versa.

Furthermore, we note that the distribution $p_{x,\beta}$ is a smooth approximation to the ideal distribution defined in Eq. 8, $p_x$, converging to it as $\beta \to 0^+$:

**Lemma 2.**

$$\lim_{\beta \to 0} p_{x,\beta} = p_x. \tag{11}$$

*(Proof in App. C).*

However, the gradient of the Reverse KL is dominated by $-\frac{1}{\beta}\nabla_\theta \mathbb{E}_{y \sim \pi_\theta}[v(y,x)]$ (Eq. 9), revealing why minimizing Reverse KL becomes an increasingly aggressive mode-seeking proxy for maximizing expected reward, at the cost of diversity even if the diversity of responses is well-captured by the target $p_x$. In the limit, with $\beta = 0$, the Reverse KL becomes undefined, the KL-Control algorithm reduces to plain REINFORCE, and no safeguard remains to preserve diversity.

## 3.2 AND BACK (AS A SPECIAL CASE OF $\alpha$-DPG)

Having defined the target distribution $p_x$, we are left with the task of picking a divergence to train a policy $\pi_\theta$ to approximate it. As we have seen, the Reverse KL $D_{\mathrm{KL}}(\pi_\theta || p)$ implicitly employed by RLVR, is a mode-seeking or zero-forcing divergence (Bishop, 2006; Huszár, 2015; Li et al., 2025). This means that it will penalize placing probability mass in regions where $p$ assigns little or none, but it is relatively indifferent to ignoring modes of $p$ altogether. As a result, the learned policy tends to concentrate on a small subset of high-probability modes while disregarding other plausible regions of the target distribution, which can reduce diversity and lead to brittle or degenerate behavior.

In contrast, the Forward KL, $D_{\mathrm{KL}}(p \,\|\, \pi_\theta)$, is mass-covering. Here, the divergence becomes large whenever $\pi_\theta$ assigns insufficient probability to regions where $p$ has support, encouraging the policy to cover all modes of the target distribution. While this can improve diversity and robustness, it often comes at the cost of assigning non-negligible probability mass to low-reward or unlikely regions, leading to less precise approximations of the target.

This tension between mode-seeking and mass-covering behavior motivates considering a broader family of divergences. The $\alpha$-divergences family (Rényi, 1961; Cressie & Read, 1984; Amari, 1985) –a subfamily of $f$-divergences– provides exactly such a continuum, interpolating smoothly between the Forward KL (as $\alpha \to 0$) and the Reverse KL (as $\alpha \to 1$). In between (for $\alpha = 0.5$) lies the squared Hellinger distance (Hellinger, 1909). By tuning $\alpha$, one can balance the degree of mode-seeking versus mass-covering behavior, potentially capturing the benefits of both extremes while mitigating their drawbacks. We refer to Table 1 for a full characterization. We denote the parametrization of $f$-DPG with $\alpha$-divergences as $\boldsymbol{\alpha}$-**DPG**, which has the following pseudo-reward:

$$R_\theta(y,x) = -f'\left(\frac{\pi_\theta(y|x)}{p_x(y)}\right) = \frac{1}{1-\alpha}\left(\left(\frac{p_x(y)}{\pi_\theta(y|x)}\right)^{1-\alpha} - 1\right). \tag{12}$$

Because for low values of $\alpha$, peaks in the $p/\pi$ ratios can induce large variance in $R_\theta$, we clip the parenthetical factor to a maximum $M$. We also rescale by discounting the constant $1/(1-\alpha)$:

$$\hat{R}_\theta(y,x) \doteq \min\left(\left(\frac{p_x(y)}{\pi_\theta(y|x)}\right)^{1-\alpha} - 1, M\right). \tag{13}$$

Finally, we use the gradient formula in Eq. 7, setting the baseline $B(x)$ to the leave-one-out per-context average of the pseudo rewards (Kool, 2019; Ahmadian et al., 2024).

It is interesting to note its behavior when setting $\alpha = 1 - \epsilon$ (i.e., close to the Reverse KL). Then,

$$\nabla_\theta D_{f_\alpha}(\pi_\theta(\cdot|x)||p_x) = \mathbb{E}_{x \sim \mathcal{D}, y \sim \pi_\theta} f'_\alpha\left(\frac{\pi_\theta(y|x)}{p_x(y)}\right)\nabla_\theta \log \pi_\theta(y|x) \tag{14}$$

$$\propto -\mathbb{E}_{x \sim \mathcal{D}, y \sim \pi_\theta}\left(\frac{p_x(y)}{\pi_\theta(y|x)}\right)^\epsilon \nabla_\theta \log \pi_\theta(y|x) \tag{15}$$

$$\approx \mathbb{E}_{x \sim \mathcal{D}, y \sim \pi_\theta} - v(y,x)\nabla_\theta \log \pi_\theta(y|x). \tag{16}$$

| Parameter | Name/Correspondence | Generator $f_\alpha(t)$ (generic) | $f'_\alpha(t)$ (generic) | $f'_\alpha(\pi/p)$ | $f'_\alpha(\infty)$ |
|---|---|---|---|---|---|
| $\alpha \neq 0, 1$ | $\alpha$–divergence ($f$-div.) | $\dfrac{t^\alpha - \alpha t - (1-\alpha)}{\alpha(\alpha-1)}$ | $\dfrac{t^{\alpha-1}-1}{\alpha-1}$ | $\dfrac{1}{\alpha-1}\left(\left(\dfrac{p}{\pi}\right)^{1-\alpha}-1\right)$ | $\alpha < 1: \frac{1}{1-\alpha}$ 
 $\alpha > 1: +\infty$ |
| $\alpha \to 1$ | **Reverse KL** $KL(\pi\|p)$ | $\lim\limits_{\alpha\to 1} f_\alpha(t) = t\log t - t + 1$ | $\lim\limits_{\alpha\to 1} f'_\alpha(t) = \log t$ | $\log(\pi/p)$ | $+\infty$ |
| $\alpha \to 0$ | **Forward KL** $KL(p\|\pi)$ | $\lim\limits_{\alpha\to 0} f_\alpha(t) = -\log t + t - 1$ | $\lim\limits_{\alpha\to 0} f'_\alpha(t) = 1 - \frac{1}{t}$ | $1 - p/\pi$ | $1$ |
| $\alpha = \frac{1}{2}$ | Hellinger (exact) | $-4\sqrt{t} + 2t + 2$ | $-\frac{2}{\sqrt{t}} + 2$ | $-2\sqrt{p/\pi} + 2$ | $2$ |

Table 1: Parametrization of the $\alpha$-divergence as an $f$-divergence $D_{f_\alpha}(\pi, p)$.

by discounting for simplicity the scaling constant and the baselines in Eq. 15, and noting that for $0 < \epsilon \ll 1$ and if $|x|$ is bounded then $x^\epsilon \approx \mathbb{I}[x \neq 0]$ and $p_x(y) \neq 0 \Leftrightarrow v(y, x) = 1$ in the last step. Thus, for $\alpha$ that is lower but very close to 1, we are again recovering the REINFORCE learning rule. In contrast, when $\alpha = 0$, then $D_{f_\alpha}$ becomes exactly the Forward KL, and thus we recover the original KL-DPG algorithm (Parshakova et al., 2019; Khalifa et al., 2021), which conserves more diversity from the original distribution sacrificing some precision:

$$\nabla_\theta D_{f_\alpha}(\pi_\theta(\cdot|x)\|p_x) = -\mathbb{E}_{x\sim\mathcal{D}, y\sim\pi_\theta}\left(\frac{p_x(y)}{\pi_\theta(y|x)} - 1\right)\nabla_\theta \log \pi_\theta(y|x) \tag{17}$$

Notably, it is easy to see that if a fixed amount of samples are obtained just from $\pi_{\text{base}}$ instead of $\pi_\theta$, KL-DPG reduces to RS-FT (Zelikman et al., 2022; Yuan et al., 2023), as it optimizes the cross-entropy to samples from the base model filtered by the verifier (see App. D for more details).

Finally, while the general formal properties of $\alpha$-divergences are well-studied, our specific setup, with a target distribution that has a restricted support (over verifiable outputs only) has some interesting novel formal properties that we describe in App. H.

## 4 EXPERIMENTS

**Informal vs Formal Mathematics**   Informal mathematics has become a common paradigm for training large language models (LLMs) on reasoning tasks, with widely used benchmarks such as MATH (Hendrycks et al., 2021) and AIME (Sun et al., 2025). These methods have achieved impressive performance (Gemini team, 2025), but they also face inherent challenges. Informal proofs and solutions often lack guarantees of rigor, making large-scale verification difficult and requiring heuristics like majority voting rather than provable correctness. While effective in many scenarios, these approaches may be less suited for tasks that aim to explore novel results or a wide variety of solutions. Formal methods provide a promising alternative (Tao, 2025). Proof assistants such as LEAN (de Moura et al., 2015), COQ (Barras et al., 1997), and ISABELLE (Nipkow et al., 2002) represent statements and proofs in a formally verifiable language. This not only allows for efficient and rigorous proof generation but also reduces dependence on labeled data. At inference, the paradigm shifts from achieving consensus over a large pool of candidates (eg., via majority voting) to ensuring sufficient diversity among them to guarantee larger coverage. Theorem provers must therefore go beyond producing a single correct proof but should generate diverse candidate proofs to more fully explore the solution space. Prior work in LEAN has applied reinforcement learning (e.g., GRPO (Wang et al., 2025; Xin et al., 2025; Ren et al., 2025)), but such methods can result in decreased diversity (He et al., 2025). In this work, we present an approach to formal theorem proving in LEAN that seeks to improve both precision for better efficiency and diversity to guarantee high coverage, highlighting the potential of formal reasoning for scalable discovery.

**Models**   For these experiments we consider DeepSeek-Prover-V1.5-SFT (Xin et al., 2025) a 7B parameters models based on DeepSeek model and further pre-trained on high-quality mathematics and code data, with a focus on formal languages such as LEAN, ISABELLE, and METAMATH, and finetuned on LEAN4 code completion datasets (Xin et al., 2025).

**Dataset**   We follow the experimental setup introduced in prior work (He et al., 2025). The training set is composed of 10K solvable LEAN problems extracted and filtered from the LEAN Workbook dataset (Ying et al., 2024; Wu et al., 2024), from which 200 problems are kept unseen as a test set.

**Reward function and LEAN4 Verifier**   The reward function extracts the last LEAN4 code block in the generated sequence and verifies it automatically by the LEAN proof assistant. The sequence is

given a reward of 1 if it is verified as correct and 0 otherwise. In our experiments we used the same LEAN4 and Mathlib4 version (lean4:v4.9.0) as used in DeepSeek-ProverV1.5 (Xin et al., 2025).

**Baselines** We compare against several baselines, focusing on critic-free methods, which have become standard thanks to not requiring an additional copy of the model: **GRPO** (DeepSeek-AI et al., 2025) –for which we only consider its unbiased instantiation, Dr. GRPO (Liu et al., 2025)– and other variants with diversity-preserving regularization: **High-KL** with a strong KL penalty($\beta = 0.1$), **Rw-Ulkly** (He et al., 2025) with a rank bias promoting diversity $\beta = 0.25$, and **Pass@k training** (Chen et al., 2025b; Tang et al., 2025) directly optimizes $\mathtt{pass@}k$ via a leave-one-out advantage formulation that reduces variance. We further include in our comparison **GPG** (Chu et al., 2025), **ReMax** (Li et al., 2024) and **RLOO** (Ahmadian et al., 2024). For reference, **Base-SFT** is the model used as a base for all methods.

**Training** All trainings are done on a single node of 4xA100 with 28 CPUs dedicated for running proof assessment in parallel. Due to the high variance in LEAN compilation and verification time, the reward function is executed asynchronously. All training scripts are based on VERL (Sheng et al., 2024). By default, since we use a verifiable reward, we follow Mistral-AI et al. (2025); Liu et al. (2025) and disable advantage normalization while setting the KL divergence penalty to $\beta = 0$. For all $\alpha$-DPG training, we set the clipping value to $M = 10$, train on float16 following Qi et al. (2025), and constrain the partition function on the lower side as $Z_x \geq \epsilon$ with $\epsilon = 1e^{-4}$. The partition function is computed online using the sampled responses for each problem, thus not incurring any additional computational cost with respect to baseline methods. See App. E for more details on the estimation of Z and an ablation experiment using more samples to compute the partition function. Across all methods, we fix the maximum response length to 1024 tokens and use 512 generated sequences per step for 200 iterations ($\approx 3$ epochs). See App. F for additional details on hyperparameters.

**Evaluation Metric** To evaluate model performance on reasoning and generation tasks, we report **pass@k** (Chen et al., 2021), a standard metric widely used in code generation and problem-solving benchmarks. The metric measures the probability that at least one correct solution is found within the top-$k$ sampled outputs of the model. For a problem with $n$ generated outputs with $n \geq k$, of which $c$ are correct, an unbiased estimate of the probability that at least one of $k$ sampled outputs is correct is (Chen et al., 2021)

$$\mathtt{pass@}k = 1 - \frac{\binom{n-c}{k}}{\binom{n}{k}}.$$

Averaging over problems gives the overall $\mathtt{pass@}k$. We generate responses using temperature $T = 1$ and nucleus sampling (Holtzman et al., 2020) with parameter $p = 0.99$.

## 4.1 RESULTS

**Coverage vs. precision analysis** We analyze models along the trade-off between coverage, measured by $\mathtt{pass@256}$, and precision, measured by $\mathtt{pass@1}$ (Fig. 1). The results reveal a clear precision–coverage frontier. The base SFT model attains relatively broad coverage but low precision, indicating the ability to solve diverse problems but with a low single-sample success probability, placing it far from the frontier. Pass@k training improves coverage while leaving precision largely unchanged. Training strategies such as GRPO with KL regularization and GRPO-Rw-Ulkly improve precision without substantially affecting coverage, still shifting performance toward a more favorable region of the trade-off space. In contrast, RLOO, GRPO without KL regularization, and GPG achieve high precision at the cost of reduced coverage. Across $\alpha$-DPG variants, lower-$\alpha$ settings (e.g., $\alpha = 0.25$) achieve the highest coverage while still considerably improving precision over SFT. Increasing $\alpha$ further improves precision, reaching parity with RL-based methods at large values (e.g., $\alpha \geq 0.995$), while typically retaining higher coverage. Importantly, most $\alpha$-DPG models lie on or near the Pareto frontier, demonstrating that the method provides a controllable and efficient trade-off between precision and coverage, outperforming or matching competing RL baselines across much of the spectrum.

**Pass@$k$ curves** Figure 2 illustrates the $\mathtt{pass@}k$ performance on the test set measured over $n = 256$ samples for a few chosen models. First, looking at the left panel, we note that we have re-

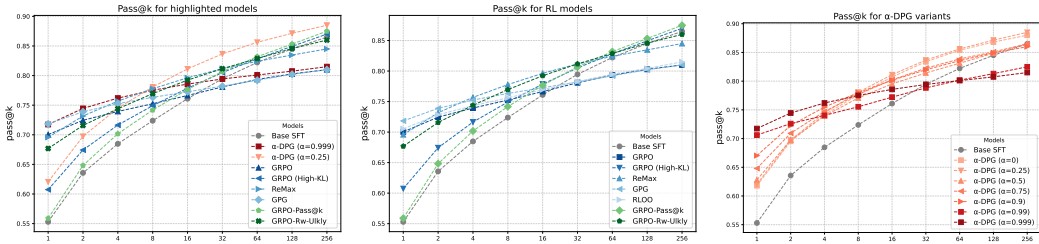

Figure 2: Pass@$k$ curves on the test set for the Base-SFT model tuned with different methods.

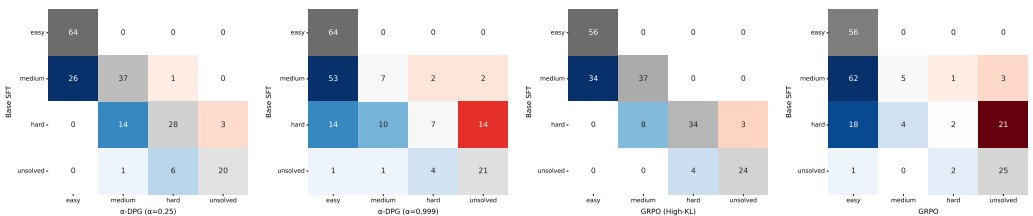

Figure 3: Problem Difficulty Transition Matrices showing the number of problems that transition from an initial difficulty classification under the base model (Base-SFT) (y-axis) to a final difficulty after post-training (x-axis). The results highlight a polarizing effect: $\alpha$-DPG ($\alpha = 0.999$) and GRPO exhibit similar behavior, improving performance on a majority of medium-difficulty problems by making them easy, but also degrading performance on hard problems, causing many of them to become unsolved. $\alpha$-DPG ($\alpha = 0.25$) and GRPO (High-KL) are more conservative, improving sample efficiency on fewer problems but harder problems remain solvable.

produced the results reported by He et al. (2025), where the GRPO model starts with a much higher pass@1 score, but then the base model overpasses it as it reaches pass@16. Furthermore, there is no single model that dominates all others across all values of $k$ indicating a Pareto trade-off between precision (pass@1) and coverage (pass@256). Nonetheless, $\alpha$-DPG ($\alpha = 0.999$) generally dominates GRPO and other pure RL-based techniques, with the only exception of ReMax that has better coverage albeit somewhat lower precision. On the other hand, $\alpha = 0.25$ dominates the base model and other diversity-preserving baselines such as Pass@k training and GRPO with KL regularization, achieving the best performance on pass@256. Rw-Ulkly, in turn, starts with higher pass@1, but $\alpha = 0.25$ outperforms for $k \geq 4$. The middle panel highlights the RL-based baselines, confirming that KL regularization helps prevent diversity collapse and Pass@k training and Rw-Ulkly baselines offer competitive results. The rightmost panel is comparing different variants of $\alpha$-DPG in relation to the base model. While the base model surpasses and matches models with $\alpha \geq 0.75$, models with lower values of alpha dominate the base model.

**Problem difficulty analysis** In this section, we examine how training affects problem solvability. We categorize problem difficulty based on model performance, measured as the proportion of correctly solved sequences. A problem is considered easy for a given model if at least 80% of the sampled sequences are correct, medium if 20–80% are correct, and hard if fewer than 20% are correct. This notion of difficulty has a direct relation with the efficiency of the model at solving a given problem as the number of samples until generating one correct solution follows a geometric distribution whose parameter is the model's sampling accuracy. In Figure 3 we plot how the problems difficulties evolve after having trained the Base SFT model using various methods. Problems on the diagonal (grey) remain unaffected by training. Elements in the lower-left triangle (blue) represent problems for which solving efficiency improved, while elements in the upper-right triangle (red) indicate problems where sampling efficiency decreased. As we can see, many problems that were medium or hard for the base model became easy after training both for GRPO and $\alpha = 0.999$. However, this came at the cost of other problems in these same categories becoming unsolvable given the same sampling budget. $\alpha$-DGP ($\alpha = 0.25$) and GRPO (High-KL) on the other hand, improve sample efficiency on fewer problems at the cost of just three problems becoming unsolvable.

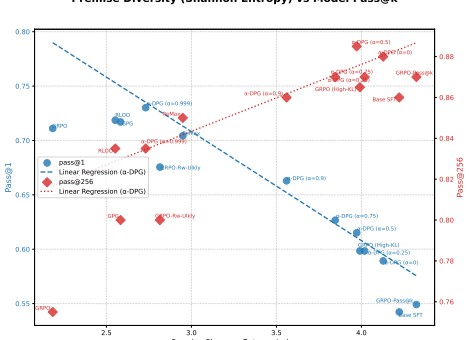
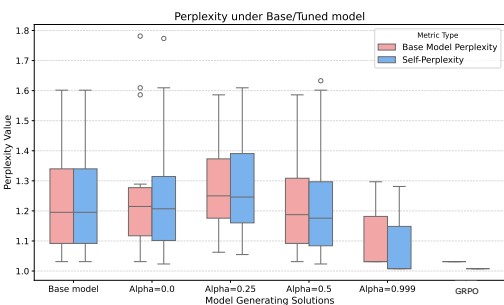

Figure 4: Left: Relationship between premise diversity measured by Shannon index and model performance (pass@1 and pass@256). The regression lines are computed for $\alpha$-DPG models. The left y-axis shows pass@1 performance, and the right y-axis shows pass@256 performance. Right: Perplexity analysis showing the distribution of perplexity for responses to a single problem sampled from various models under the base SFT model distribution.

**Diversity Analysis** We investigate the relationship between proof diversity and model performance. We split this analysis into two components: the *tactics* and *premises* used in candidate LEAN proofs. A *tactic* is a command that transforms a proof goal into simpler subgoals (e.g., `intro`, `apply`, `rw`), while a *premise* is a lemma or previously proven theorem that can be used within a proof (e.g., `mul_comm`, `mul_assoc`). As measures of diversity, we employ the Shannon index and the Gini-Simpson index. For each problem, we evaluate 256 generated proof sequences. At every proof state, we compute both the Simpson index and Shannon entropy over the choices of tactics and premises. Concretely, for a given problem, we count the occurrences of each premise and tactic, and compute the Simpson index as $D = 1 - \sum_{i=1}^{S} p_i^2$ and the Shannon index as $H = -\sum_{i=1}^{S} p_i \ln p_i$ where $p_i$ is the relative abundance of premise or tactic $i$, and $S$ is the total number of premises or tactics. These metrics are then aggregated across all problems to capture the overall diversity of candidate sequences. Higher diversity in tactics and premises in candidate proofs generally correlates with improved `pass@256` performance, whereas it is anticorrelated with `pass@1` as shown on the left panel of Figure 4 (and, additionally, in Appendix Figure 10).

**Perplexity Analysis** Recent work on RLVR (Yue et al., 2025a) shows that RL-trained models do not truly discover new solutions, instead the solutions they generate are already likely under the base model. To further investigate whether $\alpha$-DPG stays close to the base model, we conduct a perplexity analysis. We sample a single problem from the test set and have each model generate 16 solutions. For each solution, we compute perplexity both under the model that generated it (self-perplexity) and under the base model (Base SFT). As shown in the right panel of Figure 4, across all models, the generated sequences are already highly probable according to the base model, with very similar perplexities. Note also that for this particular problem, GRPO collapsed and produces 16 identical sequences, which were also highly probable under the base model.

## 5 OTHER RELATED WORK

**Improving test-time scaling (`pass@k`)** Popular RL-based post-training methods, such as PPO (Schulman et al., 2017) or GRPO (Shao et al., 2024) optimize inherently mode-seeking objectives, which can lead to mode collapse. The resulting models often achieve high accuracy, but exhibit very low entropy, generating less diverse sequences (Yue et al., 2025b). This issue is particularly pronounced when scaling test-time compute for solution search, with the SFT models often achieving better performances due to higher diversity (He et al., 2025; Chen et al., 2025a; Tang et al., 2025; Zhu et al., 2025; Yue et al., 2025a). Multiple approaches try to overcome this issue, mainly by adapting the advantage function. (He et al., 2025) proposed to add a rank bias penalty, that increases the advantage of unlikely sequences. (Chen et al., 2025b) modify the reward from `pass@1` equivalent to `pass@k`, allowing better sampling efficiency. They obtain a better `pass@k`

at inference than training with entropy regularization. Tang et al. (2025) propose a leave one out strategy to reduce variance and therefore improving pass@k at test-time.

**Reinforcement Learning from Proof Assistant Feedback** Significant efforts in the community have focused on training LLMs integrated with interactive proof assistants such as LEAN (de Moura et al., 2015), COQ (Barras et al., 1997), and ISABELLE (Nipkow et al., 2002). Early approaches leveraged LLMs to generate the next proof step or tactic (AlphaProof and AlphaGeometry teams, 2024; Polu et al., 2023; Wu et al., 2024), often combined with explicit search strategies (Lample et al., 2022). More recent work has shifted towards training models to generate complete proofs directly (Xin et al., 2025; Ren et al., 2025; Wang et al., 2025) where the last training stage relies on reinforcement learning from proof assistant feedback (RLPAF), where the proof assistant verification serves as a reward signal. However, RLPAF algorithms such as GRPO are mode-seeking, strongly truncating/filtering the original distribution, which inherently limits the diversity of generated proofs and introduces significant inefficiencies in inference scaling (Zhu et al., 2025).

## 6 FINAL REMARKS AND CONCLUSIONS

We have introduced DMVR, a general framework for optimizing a policy to produce only correct answers according to a verifier function. This perspective casts RLVR training in a new light and helps diagnose its failure modes. In particular, building on the work of Korbak et al. (2022b), we established that RLVR methods optimize toward a *filtered* version of the original distribution, even if they do it in a way that especially focuses on certain regions of high verifier reward. From this viewpoint, we can revisit recent debates on whether RL alone can create new skills (DeepSeek-AI et al., 2025; He et al., 2025; Yue et al., 2025b; Wu et al., 2025) and understand why RLVR does not generate fundamentally new capabilities but instead reweights and amplifies behaviors already present in the base model. Moreover, because RLVR is tied to a mode-seeking divergence, it sacrifices distributional breadth, leading models to forget solutions that the base model could originally provide.

However, the core principle of filtering the base model is sound and can be independently motivated; it enforces correctness while preserving the multiplicity of valid responses. The true source of diversity loss, therefore, lies not in the target distribution itself, but in the divergence used to approximate it with gradient descent within a restricted parametric family.

By explicitly defining the target distribution, DMVR enables optimization with divergences that balance the two competing goals: correctness and diversity. In particular, we explored $\alpha$-divergences, which smoothly interpolate between Forward and Reverse KL. This approach generates a Pareto frontier of models: setting $\alpha$ near the Reverse KL recovers models that match or exceed the performance of RL-based alternatives, while low values of $\alpha$ produce models that preserve substantial diversity while still improving sampling precision over the base model.

The choice of divergence impacts the resulting models in at least two different ways: On one hand, loss landscapes associated with different divergences can produce different training dynamics. On the other hand, within a restricted parametric family, each divergence can induce different optima. To disentangle the contribution that each of these factors has in the resulting models we could adopt a *curriculum* in which we gradually increase $\alpha$ during training, thus encouraging coverage at the beginning of training, and precision towards the end. If the resulting models preserve more diversity, this could be an indication that training dynamics matter. If, on the contrary, they reach as much coverage as using a high value of $\alpha$ from the start, then this would be an indication that it is the optima for the given parametric family that dominates. We leave these questions for future work.

**Known Limitations** Without clipping, $\alpha$-DPG is unstable for small values of $\alpha$ (e.g., $\leq 0.5$). Khalifa et al. (2021) use an offline version of DPG where the sampling policy is only updated from the training policy if the divergence to the target probability is estimated to have improved. Here, we avoided keeping in memory a second copy of the model, and preferred to manage variance by clipping the pseudo rewards. Also, the evidence for the effectiveness of $\alpha$-DPG centers on the Lean task using the DeepSeek Prover base model. Generalization to other tasks (e.g., code generation) and larger model families remains limited and is acknowledged as future work.

ETHICS STATEMENT

LLMs tuned using verifier feedback have attracted considerable attention from the research community and the general public because of their strong problem-solving abilities. The techniques introduced in this paper aim to preserve more of a model's initial diversity than existing RL-based approaches at the cost of less strict adherence to the verifier's feedback.

In the specific setting we study—training models to prove theorems—this trade-off carries little risk of harm. In broader applications, however, such as attempts to "align" language models with specific policies or behaviors, weaker adherence to constraints could be more concerning. We note that recent proposals within the distributional matching framework may help address this issue (Kim et al., 2025), even though they rely on the availability of a verifier at inference time.

More importantly, the distributional perspective makes explicit the central choice of a target distribution to optimize. Although the training techniques used for approximation also influence model behavior, we believe that making the choice of target distribution open and transparent can promote accountability and clarity. We therefore encourage this practice when tuning models for specific goals.

REPRODUCIBILITY STATEMENT

We will make publicly available all the code to reproduce our experiments after publication. The LEAN Workbook dataset we used in our experiments is already open (Ying et al., 2024; Wu et al., 2024), as so it is the base model DeepSeek-Prover-V1.5-SFT we use for training (Xin et al., 2025). In addition to the experiment details we describe in Section 4, we report additional details such as hyperparameter choices in in App. F.

ACKNOWLEDGEMENTS

We thank Thibaut Thonet for the very helpful comments on this paper and the anonymous reviewers for suggestions that helped improve this work.

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

## A  LLM USAGE

We have required the assistance from LLMs (ChatGPT, Gemini) in the production of this paper on many stages during its development. The most prevalent usage has been on writing code and debugging, but we have also discussed research directions and mathematical statements, used them to simplify and format mathematical proofs, proof-reading of the paper and improving the flow of the text, and they even assisted us in producing the cartoon illustrating our method. The authors take full responsibility for the contents in this paper.

## B  PROOF OF THE CORRESPONDENCE BETWEEN RLVR AND REVERSE KL

We work in the contextual setting where contexts $x$ are drawn from $\mu(x)$, the policy is $\pi_\theta(y \mid x)$, the reward is $v(y, x)$, and the prior (reference policy) is $\pi_{\text{ref}}(y \mid x)$. Define for each context $x$ and inverse temperature $\beta > 0$

$$p_{x,\beta}(y) \;=\; \frac{1}{Z_x(\beta)}\,\pi_{\text{ref}}(y \mid x)\,\exp\big(v(y,x)/\beta\big), \qquad Z_x(\beta) \;=\; \sum_y \pi_{\text{ref}}(y \mid x)\,\exp\big(v(y,x)/\beta\big).$$

Fix a context $x$. The reverse KL divergence (from $\pi_\theta(\cdot \mid x)$ to $p_{x,\beta}$) is

$$\text{KL}\big(\pi_\theta(\cdot \mid x) \,\big\|\, p_{x,\beta}\big) = \mathbb{E}_{y\sim\pi_\theta(\cdot\mid x)}\left[\log \frac{\pi_\theta(y \mid x)}{p_{x,\beta}(y)}\right].$$

Substituting $p_{x,\beta}(y) = \frac{1}{Z_x(\beta)}\pi_{\text{ref}}(y \mid x)e^{v(y,x)/\beta}$ gives

$$\text{KL}\big(\pi_\theta\|p_{x,\beta}\big) = \mathbb{E}_{y\sim\pi_\theta}\left[\log \pi_\theta(y \mid x) - \log\left(\tfrac{1}{Z_x(\beta)}\pi_{\text{ref}}(y \mid x)e^{v(y,x)/\beta}\right)\right]$$

$$= \mathbb{E}_{y\sim\pi_\theta}\left[\log \frac{\pi_\theta(y \mid x)}{\pi_{\text{ref}}(y \mid x)} - \frac{1}{\beta}v(y,x)\right] + \log Z_x(\beta).$$

Equivalently (rearranging signs),

$$\text{KL}\big(\pi_\theta\|p_{x,\beta}\big) = -\,\mathbb{E}_{y\sim\pi_\theta}\left[\frac{1}{\beta}v(y,x) - \log \frac{\pi_\theta(y \mid x)}{\pi_{\text{ref}}(y \mid x)}\right] + \log Z_x(\beta).$$

Now take the gradient with respect to $\theta$ and average over contexts $x \sim \mu$. Since $Z_x(\beta)$ depends only on $\pi_{\text{ref}}$ and $v$ (not on $\theta$), $\nabla_\theta \log Z_x(\beta) = 0$. Thus

$$\nabla_\theta \mathbb{E}_{x \sim \mu}\big[\text{KL}(\pi_\theta \| p_{x,\beta})\big] = -\nabla_\theta \mathbb{E}_{x \sim \mu}\, \mathbb{E}_{y \sim \pi_\theta(\cdot|x)}\left[\frac{1}{\beta} v(y,x) - \log \frac{\pi_\theta(y \mid x)}{\pi_{\text{ref}}(y \mid x)}\right].$$

To proceed we use the score-function (REINFORCE) identity: for any function $h(y,x)$ independent of $\theta$,

$$\nabla_\theta \mathbb{E}_{y \sim \pi_\theta(\cdot|x)}[h(y,x)] = \mathbb{E}_{y \sim \pi_\theta(\cdot|x)}\big[h(y,x)\,\nabla_\theta \log \pi_\theta(y \mid x)\big].$$

Applying this identity (and noting the dependence of the $\log \pi_\theta$ term must be handled consistently) yields the explicit gradient form

$$\nabla_\theta \mathbb{E}_{x \sim \mu}\big[\text{KL}(\pi_\theta \| p_{x,\beta})\big] = -\mathbb{E}_{x \sim \mu}\, \mathbb{E}_{y \sim \pi_\theta(\cdot|x)}\left[\left(\frac{1}{\beta} v(y,x) - \log \frac{\pi_\theta(y \mid x)}{\pi_{\text{ref}}(y \mid x)}\right)\nabla_\theta \log \pi_\theta(y \mid x)\right]. \tag{18}$$

This is the standard policy-gradient / score-function expression for the gradient of the reverse-KL objective in the contextual case.

Finally, an equivalent compact form is obtained by moving the $\nabla_\theta$ inside the expectation and noticing the factor $1/\beta$:

$$\nabla_\theta \mathbb{E}_x\big[\text{KL}(\pi_\theta \| p_{x,\beta})\big] = -\frac{1}{\beta}\nabla_\theta \mathbb{E}_{x \sim \mu,\, y \sim \pi_\theta}\left[v(y,x) - \beta \log \frac{\pi_\theta(y \mid x)}{\pi_{\text{ref}}(y \mid x)}\right].$$

$\square$

Maximizing the expected reward under the conditional policy:

$$\mathbb{E}_{x,y \sim \pi_\theta}[v(y,x) - \beta \log \frac{\pi_\theta(y \mid x)}{\pi_{\text{ref}}(y \mid x)}]$$

is equivalent (up to constants) to minimizing:

$$\mathbb{E}_{x \sim \mu(x)}\left[D_{\text{KL}}(\pi_\theta(y \mid x) \| p_{x,\beta}(y \mid x))\right].$$

## C  PROOF THAT $p_\beta$ BECOMES $p$ AS $\beta \to 0$

**Proposition 1.**
$$\lim_{\beta \to 0} p_{x,\beta} = p_x, \tag{19}$$

*in the formal sense that $\|p_{x,\beta} - p_x\|_{\text{TV}} \to 0$ as $\beta \downarrow 0$. In particular $p_{x,\beta}(y) \to p_x(y)$ for every fixed $y$.*

*Proof.* The proof is a direct adaptation, for a context $x$, of the following Lemma.

**Lemma 3.** *Let $\pi_{\text{base}}$ be a probability distribution on a countable set $Y$, let $r : Y \to \{0,1\}$ and assume $Z := \sum_y \pi_{\text{base}}(y)r(y) \in (0,1]$. Define*

$$p(y) = \frac{\pi_{\text{base}}(y)r(y)}{Z}, \qquad p_\beta(y) = \frac{\pi_{\text{base}}(y)e^{r(y)/\beta}}{Z_\beta}, \qquad Z_\beta := \sum_y \pi_{\text{base}}(y)e^{r(y)/\beta}.$$

*Then $\|p_\beta - p\|_{\text{TV}} \to 0$ as $\beta \downarrow 0$. In particular $p_\beta(y) \to p(y)$ for every fixed $y \in Y$.*

*Proof.* Set $S_0 := \sum_{r(y)=0} \pi_{\text{base}}(y) = 1 - Z$ and $\varepsilon := e^{-1/\beta}S_0 = e^{-1/\beta}(1 - Z)$. A short computation gives $Z_\beta = e^{1/\beta}Z + S_0$ and hence, after multiplying numerator and denominator by $e^{-1/\beta}$,

$$p_\beta(y) = \begin{cases} \dfrac{\pi_{\text{base}}(y)}{Z + \varepsilon}, & r(y) = 1, \\[2mm] \dfrac{e^{-1/\beta}\,\pi_{\text{base}}(y)}{Z + \varepsilon}, & r(y) = 0. \end{cases}$$

Therefore

$$\sum_{r(y)=1} \left| p_\beta(y) - p(y) \right| = \frac{\varepsilon}{Z+\varepsilon}, \qquad \sum_{r(y)=0} \left| p_\beta(y) - p(y) \right| = \frac{\varepsilon}{Z+\varepsilon},$$

so $\sum_y |p_\beta(y) - p(y)| = \dfrac{2\varepsilon}{Z+\varepsilon}$ and

$$\|p_\beta - p\|_{\mathrm{TV}} = \frac{1}{2}\sum_y |p_\beta(y) - p(y)| = \frac{\varepsilon}{Z+\varepsilon} = \frac{e^{-1/\beta}(1-Z)}{Z + e^{-1/\beta}(1-Z)}.$$

Since $e^{-1/\beta} \to 0$ as $\beta \downarrow 0$, the right-hand side tends to 0, proving total-variation convergence. The pointwise convergence $p_\beta(y) \to p(y)$ follows immediately because $|p_\beta(y)-p(y)| \le \sum_{y'} |p_\beta(y') - p(y')| = 2\|p_\beta - p\|_{\mathrm{TV}}$. $\qquad\square$

## D   RS-FT OPTIMIZES THE FORWARD KL

Rejection sampling fine tuning (Zelikman et al., 2022; Yuan et al., 2023) generates a sample set from the base model $\pi_{\mathrm{base}}$, $\mathcal{S} \sim \pi_{\mathrm{base}}(\cdot|x)$, filters them using the verifier to obtain $\mathcal{S}' = \{y_i : v(y_i, x) = 1, y_i \in \mathcal{S}\}$, and then trains the policy using standard cross-entropy on this filtered set:

$$\nabla_\theta \mathcal{L}^{\mathrm{RS\text{-}FT}} = -\mathbb{E}_{y \sim \mathcal{S}'} \nabla_\theta \log \pi_\theta(y|x) \tag{20}$$

Now, starting from KL-DPG, we note that:

$$\nabla_\theta D_{\mathrm{KL}}(p_x \| \pi_\theta) = -\mathbb{E}_{y\sim\pi_\theta(\cdot|x)} \left( \frac{p_x(y)}{\pi_\theta(y|x)} - 1 \right) \nabla_\theta \log \pi_\theta(y|x). \tag{21}$$

$$= -\mathbb{E}_{y\sim\pi_\theta(\cdot|x)} \frac{p_x(y)}{\pi_\theta(y|x)} \nabla_\theta \log \pi_\theta(y|x). \tag{22}$$

$$= -\mathbb{E}_{y\sim\pi_\theta(\cdot|x)} \frac{\pi_{\mathrm{base}}(y|x)}{\pi_{\mathrm{base}}(y|x)} \frac{p_x(y)}{\pi_\theta(y|x)} \nabla_\theta \log \pi_\theta(y|x) \tag{23}$$

$$= -\mathbb{E}_{y\sim\pi_{\mathrm{base}}(\cdot|x)} \frac{p_x(y)}{\pi_{\mathrm{base}}(y|x)} \nabla_\theta \log \pi_\theta(y|x) \tag{24}$$

$$= -\frac{1}{Z_x} \mathbb{E}_{y\sim\pi_{\mathrm{base}}(\cdot|x)} \frac{\pi_{\mathrm{base}}(y|x) v(y,x)}{\pi_{\mathrm{base}}(y|x)} \nabla_\theta \log \pi_\theta(y|x) \tag{25}$$

$$\propto \mathbb{E}_{y\sim\pi_{\mathrm{base}}(\cdot|x)} v(y,x) \nabla_\theta \log \pi_\theta(y|x), \tag{26}$$

which exactly corresponds to the objective in Eq. 20. There are two crucial differences between the two: One is their sample efficiency: whereas RS-FT is limited to using samples from the base model, many of which may be rejected from the verifier, KL-DPG makes use of the updated policy which has a higher acceptance rate. The second one is that whereas RS-FT uses a finite pool of samples, and thus can over-fit to them, KL-DPG uses an unbounded number of samples.

## E   ESTIMATION OF THE PARTITION FUNCTION

Define

$$P_x(y) = \pi_{\mathrm{base}}(y|x) v(y,x). \tag{27}$$

over a discrete space $\mathcal{Y}$, with partition function $Z_x$

$$Z_x = \sum_{y\in\mathcal{Y}} P_x(y). \tag{28}$$

Using importance sampling (Owen, 2013) we can estimate $Z_x$ by using $N$ samples $[y_i]_{i \in \{0...N\}}$ generated from a proposal distribution $q$ with $\text{Supp}(P) \subseteq \text{Supp}(q)$, as follows.

$$Z_x = \sum_{y \in \mathcal{Y}} P_x(y) = \sum_{y \in \mathcal{Y}} \frac{q(y|x)}{q(y|x)} P_x(y) = \mathbb{E}_{y \sim q(\cdot|x)} \frac{P_x(y)}{q(y|x)} \approx \frac{1}{N} \sum_i \frac{P_x(y_i)}{q(y_i|x)}. \tag{29}$$

Note that in the specific case that use $q = \pi_{\text{base}}$, then

$$Z_x = \mathbb{E}_{y \sim q(\cdot|x)} \frac{\pi_{\text{base}}(y|x)v(y,x)}{\pi_{\text{base}}(y|x)} = \mathbb{E}_{y \sim q(\cdot|x)} v(y,x) \approx \frac{1}{N} \sum_i v(y_i, x). \tag{30}$$

### E.1 ABLATION EXPERIMENT: ROLE OF THE PARTITION FUNCTION CALCULATION

To analyze whether a more precise calculation of the partition function affects the results, we compared the reported results using an online-computed partition function based on just 4 sampled responses with a pre-computed partition function using 128 samples from the base model. The results are displayed on Fig. 5, showing no clear advantage for pre-computing the partition function. As a result, we favored the online computation as it offers a drop-in replacement for GRPO and other similar variants without any additional computational burden.

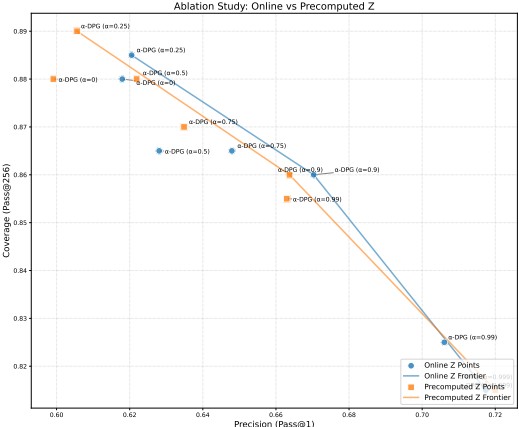

Figure 5: Comparison of $\alpha$-DPG with $Z_x$ computed online on the basis of just 4 samples per problem versus using 128 samples to compute the partition function offline.

## F ADDITIONAL EXPERIMENTAL DETAILS AND HYPERPARAMETERS

| Method | Learning Rate | Batch Size | Rollout Size (N) | KL Penalty |
|---|---|---|---|---|
| Pass@k | $1 \times 10^{-6}$ | 16 | 32 | 0.001 |
| Rwrd. Unlkly (rank plty=0.25) | $2 \times 10^{-6}$ | 16 | 32 | 0.001 |
| Dr. GRPO | $2 \times 10^{-6}$ | 128 | 4 | 0.0 |
| RLOO | $2 \times 10^{-6}$ | 128 | 4 | 0.0 |
| GPG | $2 \times 10^{-6}$ | 128 | 4 | 0.0 |
| ReMax | $2 \times 10^{-6}$ | 128 | 4 | 0.0 |
| Dr. GRPO with High KL | $2 \times 10^{-6}$ | 128 | 4 | 0.1 |
| $\alpha$-DPG | $2 \times 10^{-6}$ | 128 | 4 | - |

Table 2: Summary of key hyper parameters for different training runs.

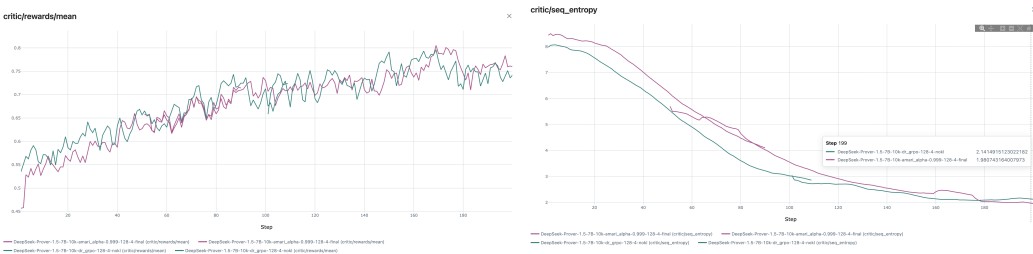

Figure 6: Training curves of both $\alpha$-DPG and dr-GRPO. Sequence entropy on the right and reward on the left

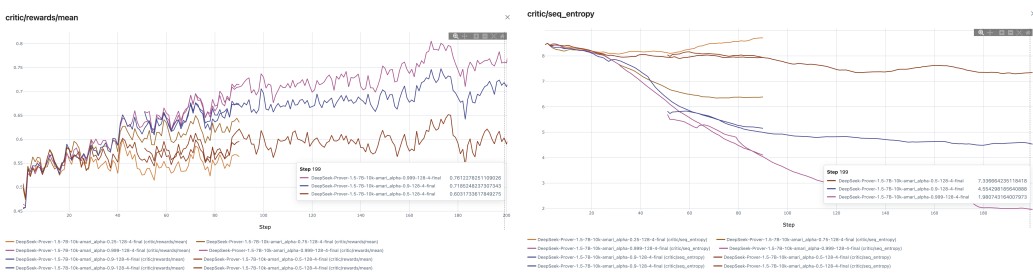

Figure 7: Training curves of $\alpha$-DPG for various alpha values. Sequence entropy on the right and reward on the left. (Truncated curves are runs that have been stopped and resumed )

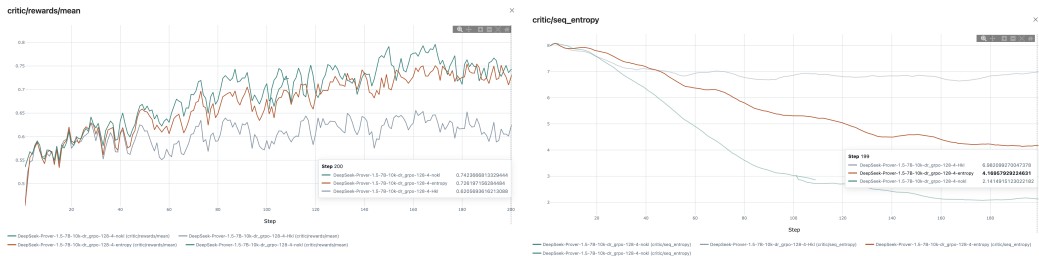

Figure 8: Training curves various dr-grpo baselines (dr-grpo, +high KL, +entropy). Sequence entropy on the right and reward on the left. (Truncated curves are runs that have been stopped and resumed )

# G  ADDITIONAL EXPERIMENTAL RESULTS

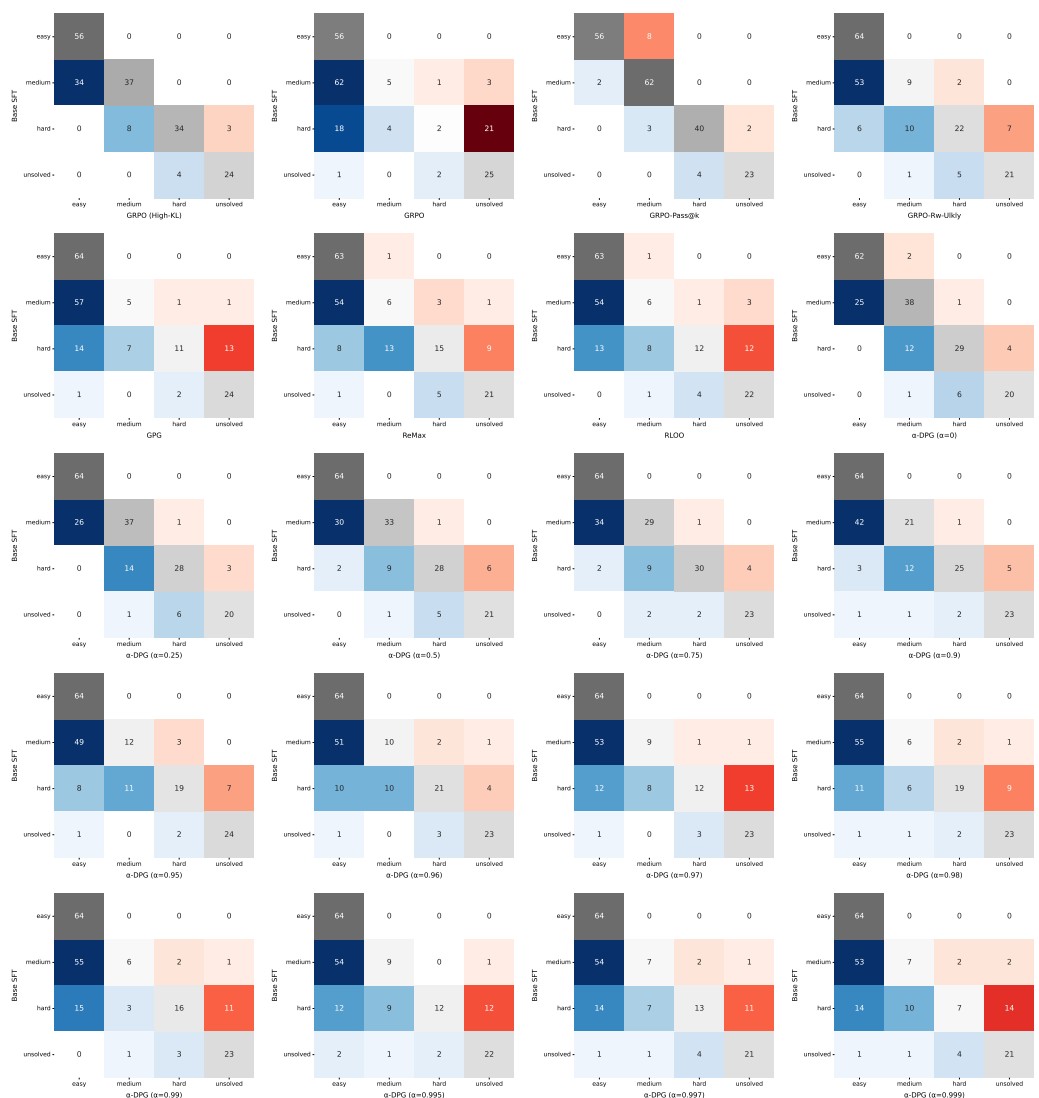

Figure 9: Problem Difficulty Transition Matrix from the Base-SFT to GRPO. The matrix shows the number of problems that transition from an initial difficulty classification under the base model (Base-SFT) (y-axis) to a final classification after post-training (x-axis).

| | Base SFT | GRPO-Pass@k | GRPO | GRPO (High-KL) | GRPO-Rw-Ulkly | ReMax | RLOO | GPG | α-DPG (α=0) | α-DPG (α=0.25) | α-DPG (α=0.5) | α-DPG (α=0.75) | α-DPG (α=0.9) | α-DPG (α=0.95) | α-DPG (α=0.96) | α-DPG (α=0.97) | α-DPG (α=0.98) | α-DPG (α=0.99) | α-DPG (α=0.995) | α-DPG (α=0.997) | α-DPG (α=0.999) |
|---|---|---|---|---|---|---|---|---|---|---|---|---|---|---|---|---|---|---|---|---|---|
| Base SFT | **55.3/86.5** | -1.0 | **+5.5** | -0.5 | +0.5 | +2.0 | **+5.0** | **+5.5** | -1.5 | -2.0 | -1.0 | +0.0 | +0.0 | +0.5 | +2.0 | +0.5 | **+5.0** | +3.0 | +4.0 | +4.0 | +3.0 | **+5.0** |
| GRPO-Pass@k | +0.6 | **55.9/87.5** | **+6.5** | +0.5 | +1.5 | **+3.0** | **+6.0** | **+6.5** | -0.5 | -1.0 | +1.0 | +1.0 | +1.5 | **+3.0** | +1.5 | **+6.0** | **+4.0** | **+5.0** | **+5.0** | +4.0 | **+6.0** |
| GRPO | **+14.7** | **+14.1** | **70.0/81.0** | **-6.0** | **-5.0** | -3.5 | -0.5 | +0.0 | **-7.0** | **-7.5** | **-5.5** | **-5.5** | **-5.0** | -3.5 | **-5.0** | -0.5 | -2.5 | -1.5 | -1.5 | -2.5 | -0.5 |
| GRPO (High-KL) | **+5.4** | **+4.8** | **-9.3** | **60.7/87.0** | +1.0 | +2.5 | **+5.5** | **+6.0** | -1.0 | -1.5 | +0.5 | +0.5 | +1.0 | +2.5 | +1.0 | **+5.5** | +3.5 | **+4.5** | **+4.5** | +3.5 | **+5.5** |
| GRPO-Rw-Ulkly | **+12.4** | **+11.8** | -2.4 | **+6.9** | **67.7/86.0** | +1.5 | **+4.5** | **+5.0** | -2.0 | -2.5 | -0.5 | -0.5 | +1.5 | -0.0 | **+4.5** | +2.5 | +3.5 | +2.5 | **+4.5** |
| ReMax | **+14.3** | **+13.7** | -0.5 | **+8.8** | +1.9 | **69.5/84.5** | **+3.0** | +3.5 | -3.5 | -4.0 | -2.0 | -2.0 | -1.5 | +0.0 | -1.5 | +3.0 | +1.0 | +2.0 | +1.0 | +3.0 |
| RLOO | **+15.2** | **+14.6** | +0.5 | **+9.8** | +2.9 | +1.0 | **70.5/81.5** | +0.5 | **-6.5** | **-7.0** | **-5.0** | **-5.0** | **-4.5** | -3.0 | **-4.5** | +0.0 | -2.0 | -1.0 | -1.0 | -2.0 | +0.0 |
| GPG | **+16.5** | **+15.9** | **+1.8** | **+11.1** | **+4.2** | **+2.3** | +1.3 | **71.8/81.0** | **-7.0** | **-7.5** | **-5.5** | **-5.5** | **-5.0** | -3.5 | **-5.0** | -0.5 | **-2.5** | -1.5 | -1.5 | -2.5 | -0.5 |
| α-DPG (α=0) | **+6.5** | **+5.9** | **-8.2** | +1.1 | **-5.9** | **-7.8** | **-8.7** | **-10.0** | **61.8/88.0** | -0.5 | +1.5 | +1.5 | +2.0 | **+3.5** | +2.0 | **+6.5** | **+4.5** | **+5.5** | **+5.5** | **+4.5** | **+6.5** |
| α-DPG (α=0.25) | **+6.7** | **+6.2** | **-8.0** | +1.3 | **-5.6** | **-7.5** | **-8.5** | **-9.8** | +0.3 | **62.0/88.5** | +2.0 | +2.0 | +2.5 | **+4.0** | **+2.5** | **+7.0** | **+5.0** | **+6.0** | **+6.0** | **+5.0** | **+7.0** |
| α-DPG (α=0.5) | **+7.5** | **+6.9** | **-7.2** | +2.1 | **-4.9** | **-6.8** | **-7.7** | **-9.0** | +1.0 | +0.8 | **62.8/86.5** | -0.0 | +0.5 | +0.5 | **+5.0** | **+3.0** | **+4.0** | **+4.0** | +3.0 | **+5.0** |
| α-DPG (α=0.75) | **+9.5** | **+8.9** | **-5.2** | +4.1 | -2.9 | -4.8 | **-5.7** | **-7.0** | +3.0 | +2.7 | **+2.0** | **64.8/86.5** | +0.5 | **+2.0** | +0.5 | **+5.0** | +3.0 | **+4.0** | **+4.0** | +3.0 | **+5.0** |
| α-DPG (α=0.9) | **+11.7** | **+11.1** | **-3.0** | **+6.3** | -0.6 | -2.5 | -3.5 | -4.8 | **+5.2** | **+5.0** | **+4.2** | **+2.2** | **67.0/86.0** | +1.5 | -0.0 | **+4.5** | +2.5 | +3.5 | +3.5 | +2.5 | **+4.5** |
| α-DPG (α=0.95) | **+13.3** | **+12.8** | -1.4 | **+7.9** | +1.0 | -0.9 | -1.9 | -3.2 | **+6.8** | **+6.6** | **+5.8** | **+3.9** | **+1.6** | **68.6/84.5** | -1.5 | +3.0 | +1.0 | +2.0 | +2.0 | +1.0 | +3.0 |
| α-DPG (α=0.96) | **+14.3** | **+13.7** | -0.4 | **+8.9** | **+2.0** | +0.1 | -0.9 | -2.2 | **+7.8** | **+7.6** | **+6.8** | **+4.8** | **+2.6** | +1.0 | **69.6/86.0** | **+4.5** | +2.5 | +3.5 | +3.5 | +2.5 | **+4.5** |
| α-DPG (α=0.97) | **+15.1** | **+14.5** | +0.3 | **+9.6** | **+2.7** | +0.8 | -0.2 | -1.5 | **+8.6** | **+8.3** | **+7.5** | **+5.6** | **+3.3** | +1.7 | +0.7 | **70.3/81.5** | -2.0 | -1.0 | -1.0 | -2.0 | +0.0 |
| α-DPG (α=0.98) | **+14.5** | **+13.9** | -0.2 | **+9.0** | **+2.1** | +0.2 | -0.7 | -2.1 | **+8.0** | **+7.7** | **+7.0** | **+5.0** | **+2.8** | +1.1 | +0.1 | -0.6 | **69.8/83.5** | -1.0 | +1.0 | -0.0 | +2.0 |
| α-DPG (α=0.99) | **+15.3** | **+14.7** | +0.3 | **+9.9** | **+2.9** | +1.0 | +0.1 | -1.2 | **+8.8** | **+8.5** | **+7.8** | **+5.8** | **+3.6** | **+2.0** | +1.0 | +0.2 | +0.8 | **70.6/82.5** | -0.0 | -1.0 | +1.0 |
| α-DPG (α=0.995) | **+16.7** | **+16.1** | +2.0 | **+11.3** | **+4.4** | **+2.5** | +1.5 | +0.2 | **+10.2** | **+10.0** | **+9.2** | **+7.3** | **+5.0** | **+3.4** | **+2.4** | +1.7 | **+2.3** | +1.4 | **72.0/82.5** | -1.0 | +1.0 |
| α-DPG (α=0.997) | **+16.9** | **+16.3** | **+2.2** | **+11.5** | **+4.6** | **+2.7** | +1.7 | +0.4 | **+10.4** | **+10.2** | **+9.4** | **+7.4** | **+5.2** | **+3.6** | **+2.6** | +1.9 | **+2.5** | +1.6 | +0.2 | **72.2/83.5** | +2.0 |
| α-DPG (α=0.999) | **+16.4** | **+15.8** | +1.7 | **+11.0** | +4.1 | **+2.2** | +1.2 | -0.1 | **+9.9** | **+9.7** | **+8.9** | **+6.9** | **+4.7** | **+3.1** | **+2.1** | +1.4 | +2.0 | +1.1 | -0.3 | -0.5 | **71.7/81.5** |

Table 3: Pairwise performance comparison. **Diagonal**: Absolute Pass@1 / Pass@256 scores (%). **Upper Triangle**: Pass@256 differences (Row − Column). **Lower Triangle**: Pass@1 differences (Row − Column). **Bold** indicates statistical significance ($p < 0.05$) via paired bootstrap.

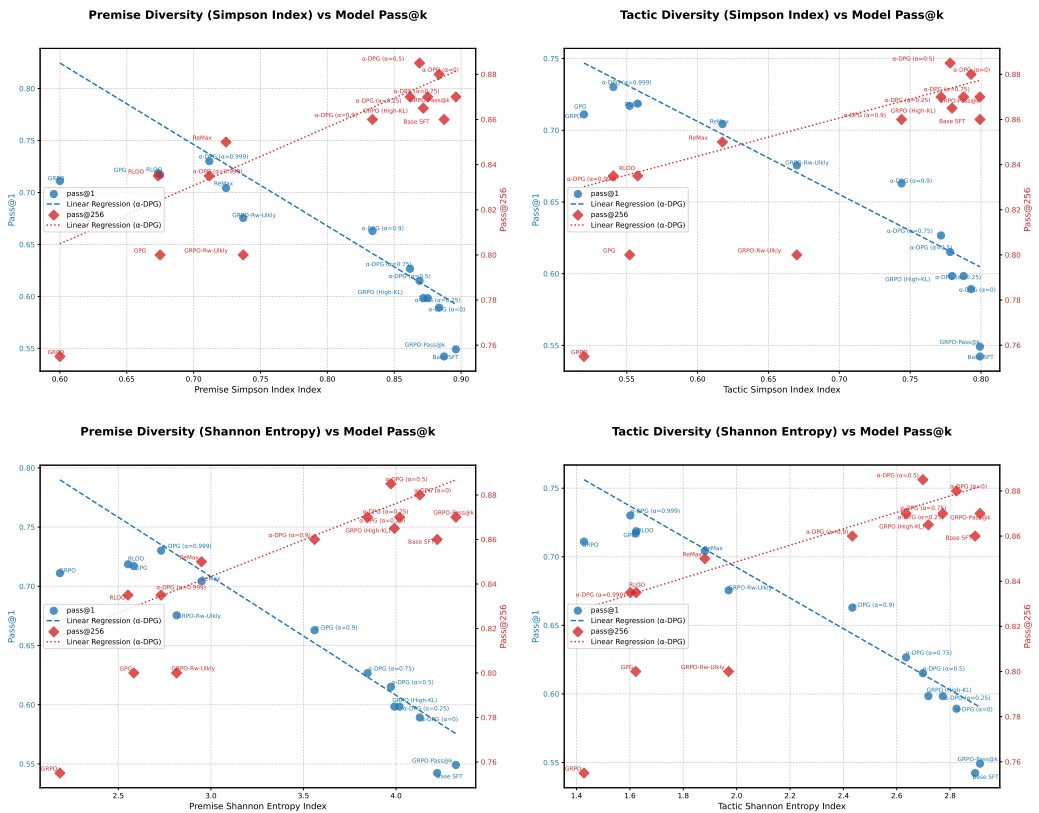

Figure 10: Diversity index vs Pass@k

| Model | Premise Simpson Index | Tactic Simpson Index | Tactic Entropy Index | Premise Entropy Index | pass@256 | pass@1 |
|---|---|---|---|---|---|---|
| Base SFT | 0.884013 | 0.793399 | 2.814039 | 4.138613 | 0.865000 | 0.553008 |
| GRPO-Pass@k | 0.893146 | 0.793183 | 2.840828 | 4.271897 | 0.875000 | 0.558867 |
| GRPO | 0.668813 | 0.558731 | 1.641007 | 2.597491 | 0.810000 | 0.700332 |
| GRPO (High-KL) | 0.867052 | 0.775942 | 2.665062 | 3.936221 | 0.870000 | 0.607383 |
| GRPO-Rw-Ulkly | 0.794354 | 0.686514 | 2.127269 | 3.231985 | 0.860000 | 0.676836 |
| ReMax | 0.712695 | 0.614204 | 1.861515 | 2.920214 | 0.845000 | 0.695762 |
| RLOO | 0.651459 | 0.557749 | 1.614147 | 2.448276 | 0.815000 | 0.705332 |
| GPG | 0.670303 | 0.550061 | 1.607887 | 2.538284 | 0.810000 | 0.718320 |
| $\alpha$-DPG ($\alpha$=0) | 0.869384 | 0.775301 | 2.636581 | 3.899748 | 0.880000 | 0.617969 |
| $\alpha$-DPG ($\alpha$=0.25) | 0.864668 | 0.776839 | 2.654524 | 3.873548 | 0.885000 | 0.620527 |
| $\alpha$-DPG ($\alpha$=0.5) | 0.864517 | 0.770372 | 2.616379 | 3.855439 | 0.865000 | 0.628066 |
| $\alpha$-DPG ($\alpha$=0.75) | 0.849480 | 0.759164 | 2.519318 | 3.678993 | 0.865000 | 0.647949 |
| $\alpha$-DPG ($\alpha$=0.9) | 0.826129 | 0.735853 | 2.344282 | 3.473220 | 0.860000 | 0.670332 |
| $\alpha$-DPG ($\alpha$=0.95) | 0.793030 | 0.699298 | 2.144001 | 3.191290 | 0.845000 | 0.686562 |
| $\alpha$-DPG ($\alpha$=0.96) | 0.792300 | 0.670038 | 2.035383 | 3.155785 | 0.860000 | 0.696621 |
| $\alpha$-DPG ($\alpha$=0.97) | 0.731704 | 0.635556 | 1.884774 | 2.848516 | 0.815000 | 0.703691 |
| $\alpha$-DPG ($\alpha$=0.98) | 0.741682 | 0.635610 | 1.877606 | 2.881272 | 0.835000 | 0.697891 |
| $\alpha$-DPG ($\alpha$=0.99) | 0.718106 | 0.584643 | 1.699220 | 2.735522 | 0.825000 | 0.706055 |
| $\alpha$-DPG ($\alpha$=0.995) | 0.683162 | 0.554934 | 1.603550 | 2.593344 | 0.825000 | 0.720449 |
| $\alpha$-DPG ($\alpha$=0.997) | 0.702118 | 0.584954 | 1.693736 | 2.687643 | 0.835000 | 0.722520 |
| $\alpha$-DPG ($\alpha$=0.999) | 0.662599 | 0.546542 | 1.573962 | 2.536636 | 0.815000 | 0.717402 |

Table 4: Diversity and performance metrics across models. For each problem statement, we evaluate 256 generated proof sequences. At every proof state, we compute the Simpson Index (SI) and Shannon entropy over both tactic choices and premise selections. The metrics are then aggregated across all problems to capture the overall diversity of candidate sequences. Higher diversity in tactics and premises generally correlates with improved pass@256 performance.

# H  FORMAL COMPLEMENTS ON $\alpha$-DIVERGENCE

## H.1  SMOOTHNESS OF $\alpha$-DIVERGENCE, BEHAVIOUR FOR $\alpha \to 0$ AND $\alpha \to 1$

The fact that $D_{f_\alpha}(\pi, p)$ is a continuous function of $\alpha$ and that it converges to the forward $KL(p||\pi)$ for $\alpha \to 0$ and to the reverse $KL(\pi||p)$ for $\alpha \to 1$ — including cases where these KL-divergences may be infinite — is well-known in the literature, e.g. Cichocki & Amari (2010).

In our specific situation, while typically the autoregressive policy $\pi$ is full-support ($\pi(y) > 0, \forall y \in \mathcal{Y}$), the support $A := \{y : p(y) > 0\}$ of $p$ is a proper subset of $\mathcal{Y}$. In such cases, while $KL(p||\pi)$ is (typically) finite,[4] $KL(\pi||p)$ is infinite.

## H.2  COMPARING DIFFERENT POLICIES, ILLUSTRATION

In order to provide a "non gradient" interpretation of what happens at the edges, it is instructive to compare $D_{f_\alpha}(\pi, p)$ with $D_{f_\alpha}(\pi', p)$ for different candidate policies $\pi$ and $\pi'$.

**Illustration**  We provide an illustration in Fig. 11, on a toy example with a small finite sample space $\mathcal{Y}$, with $A \subsetneq \mathcal{Y}$, and with three policies. The policy $\pi_1$ is more "covering" than $\pi_2$ and $\pi_3$, with a lower forward $KL(p||\pi)$, but it is less "focussed" on the valid region $A$ than either $\pi_2$ or $\pi_3$, which are both concentrated on $A$ (with $\pi_2(A) = \pi_3(A) = 0.9$), but with different peaks. Despite the fact that the reverse $KL(\pi||p)$ is infinite for the three policies, the divergences, for $\alpha$ close to 1 — while large (and tending to infinity) — still show a clear order, with $D_{f_\alpha}(\pi_1, p)$ much higher than $D_{f_\alpha}(\pi_3, p)$, which in turn is slightly higher than $D_{f_\alpha}(\pi_2, p)$.

More in detail, we consider the discrete space $\mathcal{Y} = \{y_1, y_2, y_3\}$, over which the base model $\pi_{base}$ has an (almost) uniform distribution $\pi_{base} = (0.33, 0.34, 0.33)$, and where the binary verifier $v(y)$ takes the values $(1, 0, 1)$, i.e. accepts $y_1$ and $y_3$ and rejects $y_2$. This results in the target distribution:

$$p = (0.5, 0, 0.5).$$

We study three alternative policies, with distributions:

$$\pi_1 = \pi_{base} = (0.33, 0.34, 0.33), \qquad \pi_2 = (0.8, 0.1, 0.1), \qquad \pi_3 = (\epsilon, 0.1, 1 - \epsilon),$$

taking $\epsilon = 0.01$. $\pi_1$ is the more diverse/covering of the three, but has some significant mass on the invalid point $y_2$, while $\pi_2$ and $\pi_3$ waste less mass on the invalid point, and are peaky on the first point and the third point respectively, even more so for $\pi_3$.

The endpoints recover the forward and reverse KL divergences:

$$\lim_{\alpha \to 0} D_{f_\alpha}(\pi||p) = KL(p||\pi), \qquad \lim_{\alpha \to 1} D_{f_\alpha}(\pi||p) = KL(\pi||p),$$

where we adopt the usual conventions that $KL(p||\pi) = +\infty$ whenever $p$ charges a point on which $\pi$ vanishes, and similarly for $KL(\pi||p)$.

In our setting, $p(y_2) = 0$ while each $\pi_i$ assigns positive mass to $y_2$, so $KL(\pi_i||p) = +\infty$ for $i = 1, 2, 3$.

**Numerical values.**  Table 5 reports $D_{f_\alpha}(\pi_i||p)$ for a range of representative $\alpha$ values, including the limiting cases $\alpha = 0$ and $\alpha = 1$.

**Curves as a function of $\alpha$.**  Figure 11 shows the curves $\alpha \mapsto D_{f_\alpha}(\pi_i||p)$ for $i = 1, 2, 3$ on $\alpha \in (0, 1)$.

On the left of the plot, with $\alpha$ at 0, we see that $\pi_1$, which is the more diverse/covering of the three relative to $p$, has the lowest value of forward KL, $KL(p, \pi)$, while $\pi_3$, which is even more "peaky" than $\pi_2$ has a larger forward KL. On the right of the plot, with $\alpha$ tending to 1, the divergences all tend to infinity, but their order stabilizes, with the divergence of $\pi_1$ getting and staying much larger

---

[4]In some "pathological" situations, and for an infinite sample space $\mathcal{Y}$, $KL(p||q)$ can be infinite even when the supports coincide.

| Distribution | $\alpha \to 0$ | $\alpha = 0.1$ | $\alpha = 0.5$ | $\alpha = 0.7$ | $\alpha = 0.9$ | $\alpha = 0.99$ | $\alpha \to 1$ |
|---|---|---|---|---|---|---|---|
| $\pi_1 = (0.33, 0.34, 0.33)$ | 0.4155 | 0.4522 | 0.7504 | 1.2018 | 3.4666 | 34.0658 | $\infty$ |
| $\pi_2 = (0.8, 0.1, 0.1)$ | 0.5697 | 0.5585 | 0.5758 | 0.6816 | 1.3252 | 10.3160 | $\infty$ |
| $\pi_3 = (0.01, 0.1, 0.89)$ | 1.6677 | 1.4693 | 1.0488 | 1.1827 | 1.7766 | 10.5828 | $\infty$ |

Table 5: Values of $D_{f_\alpha}(\pi_i \| p)$ for $p = (0.5, 0, 0.5)$ and $\pi_1, \pi_2, \pi_3$ at selected $\alpha$ values. For $\alpha \to 0$ and $\alpha \to 1$ we recover the forward and reverse KL divergences, respectively.

than both those of $\pi_2$ and $\pi_3$. As we will see in Theorem 5 and equation 32 below, the relative behaviour of the policies for $\alpha$ close to 1 is determined predominantly by their relative values of $\pi(A)$, and here, with $\pi_2(A) = \pi_3(A) = 0.9 > \pi_1(A) = 0.66$, $\pi_1$ "loses" relative to $\pi_2$ and $\pi_3$. For two policies, such as $\pi_2, \pi_3$, which have exactly the same "valid" mass, their order is determined by a secondary term, the one appearing to the right of '+' in equation 32.

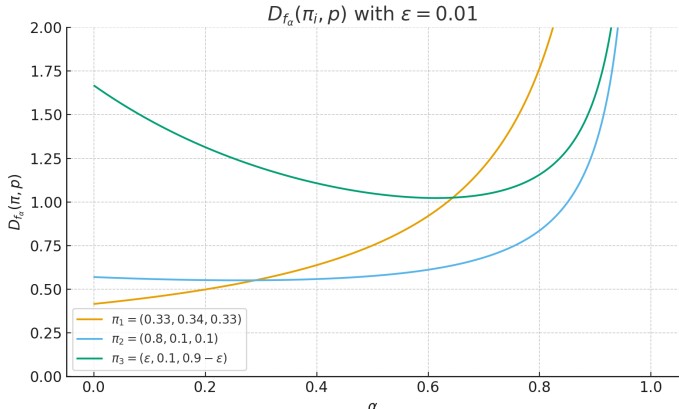

Figure 11: The divergence $D_{f_\alpha}(\pi, p)$ for $\pi_1 = (0.33, 0.34, 0.33)$, $\pi_2 = (0.8, 0.1, 0.1)$, and $\pi_3 = (\varepsilon, 0.1, 0.9 - \varepsilon)$ with $\varepsilon = 0.01$.

### H.3 A DECOMPOSITION THEOREM FOR THE $\alpha$-DIVERGENCE FOR TARGETS WITH PARTIAL SUPPORT

We now state a useful (and apparently novel) result, which permits a better understanding of what happens in the situation where the support $A$ of the target $p$ is strictly contained in the support of the model $\pi$. This result says that, with $\pi(A)$ the $\pi$ mass of $A$, and with $\pi_A$ is the "renormalization" of $\pi$ to $A$, that is, $\pi_A(y) = \pi(y)/\pi(A)$ for $y \in A$, and for $\alpha \in (0, 1)$, we have the identity $D_{f_\alpha}(\pi, p) = \frac{1 - \pi(A)^\alpha}{\alpha(1 - \alpha)} + \pi(A)^\alpha D_{f_\alpha}(\pi_A, p)$.

This identity is especially interesting for the case of $\alpha$ tending to 1. In that case, with $\pi$ full support and $\pi(A) < 1$, the support of $\pi_A$ is equal to the support of $p$, and therefore $D_{f_\alpha}(\pi_A, p)$ tends to a finite value $KL(\pi_A, p)$, and the second term of the identity tends towards a finite value. On the other hand, the first term tends to infinity at a rate closer and closer to $\frac{1 - \pi(A)}{1 - \alpha}$, meaning that $\pi(A) > \pi'(A)$ implies that the divergence of $\pi$ becomes and stays lower than the divergence of $\pi'$ after a certain point $\alpha_0$.

*In other words, when $A$ is the subset of $\mathcal{Y}$ for which the binary reward $v(y)$ is equal to 1, and for $\alpha$ sufficiently close to 1, minimizing $D_{f_\alpha}(\pi_\theta, p)$ is essentially equivalent to maximizing $\mathbb{E}_{\pi_\theta} v(y)$, the same objective as pure REINFORCE.*

### FORMAL RESULT: THE SUPPORT DECOMPOSITION OF $\alpha$-DIVERGENCE

Let $\pi$ and $p$ be probability distributions on a countable sample space $Y$. We consider the $\alpha$-divergence defined by the generator function $f_\alpha(t) = \frac{t^\alpha - \alpha t - (1 - \alpha)}{\alpha(\alpha - 1)}$ for $\alpha \in (0, 1)$.

First, we establish the algebraic relationship between the divergence and the "Hellinger sum" (the discrete counterpart to the Hellinger integral (Liese & Vajda, 2006), see also Appendix B of (Li & Gal, 2017)).

**Lemma 4** (Connection to Hellinger sum). *Let $\alpha \in (0, 1)$. Let $A = \mathrm{supp}(p) \subseteq Y$. The $\alpha$-divergence satisfies:*

$$D_{f_\alpha}(\pi, p) = \frac{1 - H_\alpha(\pi, p)}{\alpha(1 - \alpha)}, \tag{31}$$

*where $H_\alpha(\pi, p) = \sum_{y \in Y} \pi(y)^\alpha p(y)^{1-\alpha}$ is the Hellinger sum. This holds even if $\mathrm{supp}(\pi) \not\subseteq A$.*

*Proof.* We use the extended definition of $f$-divergence (Polyanskiy & Wu, 2025) which includes the boundary term for the set where $p(y) = 0$ but $\pi(y) > 0$. Let $A^c = Y \setminus A$ and let $\epsilon = \pi(A^c)$ be the "leakage" mass.

$$D_{f_\alpha}(\pi, p) = \sum_{y \in A} p(y) f_\alpha\left(\frac{\pi(y)}{p(y)}\right) + f'_\alpha(\infty) \cdot \pi(A^c).$$

1. **The Boundary Term:** The term to the right uses the constant $f'_\alpha(\infty) = \lim_{t \to \infty} f_\alpha(t)/t$. For $\alpha < 1$, $t^{\alpha-1} \to 0$, so: $f'_\alpha(\infty) = \frac{-\alpha}{\alpha(\alpha-1)} = \frac{1}{1-\alpha}$. See also Table 1.

2. **The Sum on Support $A$:** Note that $\sum_{y \in A} \pi(y) = 1 - \epsilon$ and $\sum_{y \in A} p(y) = 1$.

$$\sum_{y \in A} p(y) f_\alpha\left(\frac{\pi(y)}{p(y)}\right) = \frac{1}{\alpha(\alpha - 1)}\left[\sum_{y \in A} \pi(y)^\alpha p(y)^{1-\alpha} - \alpha(1 - \epsilon) - (1 - \alpha)(1)\right]$$

$$= \frac{1}{\alpha(\alpha - 1)}\left[H_\alpha(\pi, p) - 1 + \alpha\epsilon\right].$$

3. **Combination:** Adding the boundary term: Boundary $= \frac{\epsilon}{1-\alpha} = \frac{-\alpha\epsilon}{\alpha(\alpha-1)}$. The terms involving $\epsilon$ cancel perfectly:

$$D_{f_\alpha}(\pi, p) = \frac{H_\alpha(\pi, p) - 1 + \alpha\epsilon - \alpha\epsilon}{\alpha(\alpha - 1)} = \frac{1 - H_\alpha(\pi, p)}{\alpha(1 - \alpha)}.$$

$\square$

Using Lemma 4, we now derive the main decomposition theorem.

**Theorem 5** (Support Decomposition). *Assume that $A = \mathrm{supp}(p)$ is strictly included in $\mathrm{supp}(\pi)$. Let $\pi_A$ be the renormalization of $\pi$ on $A$, i.e., $\pi_A(y) = \pi(y)/\pi(A)$ for $y \in A$, or, equivalently $\pi_A(y) = \pi(y|A)$. The divergence decomposes as:*

$$D_{f_\alpha}(\pi, p) = \underbrace{\frac{1 - \pi(A)^\alpha}{\alpha(1 - \alpha)}}_{\text{Leakage Penalty}} + \underbrace{\pi(A)^\alpha D_{f_\alpha}(\pi_A, p)}_{\text{Shape Divergence}}. \tag{32}$$

*Proof.* Since $p(y) = 0$ for $y \notin A$, the Hellinger sum restricts to $A$. Substituting $\pi(y) = \pi(A)\pi_A(y)$:

$$H_\alpha(\pi, p) = \sum_{y \in A} (\pi(A)\pi_A(y))^\alpha p(y)^{1-\alpha} = \pi(A)^\alpha H_\alpha(\pi_A, p).$$

From Lemma 4, we have $H_\alpha(\pi_A, p) = 1 - \alpha(1 - \alpha)D_{f_\alpha}(\pi_A, p)$. Substituting this back into the global divergence formula:

$$D_{f_\alpha}(\pi, p) = \frac{1 - \pi(A)^\alpha H_\alpha(\pi_A, p)}{\alpha(1 - \alpha)}$$

$$= \frac{1 - \pi(A)^\alpha \left[1 - \alpha(1 - \alpha)D_{f_\alpha}(\pi_A, p)\right]}{\alpha(1 - \alpha)}$$

$$= \frac{1 - \pi(A)^\alpha}{\alpha(1 - \alpha)} + \pi(A)^\alpha D_{f_\alpha}(\pi_A, p).$$

$\square$

Consequences for Fixed Target $p$

We analyze the case where $\pi$ has full support and $p$ has strictly partial support $A$.

**Remark 1** (Limit $\alpha \to 1$: The Strong Constraint). *As $\alpha \to 1$, $D_{f_\alpha}$ converges to the Reverse KL divergence $D_{KL}(\pi\|p) = +\infty$. The Mass Penalty term diverges:*

$$\lim_{\alpha \to 1} \frac{1 - \pi(A)^\alpha}{\alpha(1-\alpha)} = +\infty \quad (\text{if } \pi(A) < 1).$$

*This acts as a strong constraint, heavily penalizing support leakage.*

*As for the Shape divergence term $\pi(A)^\alpha D_{f_\alpha}(\pi_A, p)$, it remains finite, and is dominated by the first term.*

**Remark 2** (Limit $\alpha \to 0$: The Weak Constraint). *As $\alpha \to 0$, $D_{f_\alpha}$ converges to the Forward KL divergence $D_{KL}(p\|\pi)$. The Mass Penalty term remains finite:*

$$\lim_{\alpha \to 0} \frac{1 - \pi(A)^\alpha}{\alpha(1-\alpha)} = -\ln(\pi(A)).$$

*This acts as a soft constraint ("Surprise Penalty"), allowing a trade-off between coverage ($\pi(A)$) and conditional shape matching ($D_{KL}(p\|\pi_A)$). The Shape divergence term $\pi(A)^\alpha D_{f_\alpha}(\pi_A, p)$ also remains finite and converges to $D_{f_\alpha}(\pi_A, p)$.*

