# OpenReview forum: "Whatever Remains Must Be True: Filtering Drives Reasoning in LLMs, Shaping Diversity"
_ICLR.cc/2026/Conference — ICLR 2026 Poster_

### Official Review · Reviewer_yHrP · 2025-10-16

**Soundness:** 3
**Presentation:** 3
**Contribution:** 2
**Rating:** 4
**Confidence:** 5

**Summary:**

This paper proposes Distributional Matching with Verifiable Rewards (DMVR), a framework for training LLMs on reasoning tasks that explicitly defines a target distribution by filtering incorrect answers while preserving relative probabilities of correct ones. The authors introduce Amari-DPG, which uses Amari's α-divergence family to interpolate between mode-seeking (Reverse KL) and mass-covering (Forward KL) divergences, enabling control over the precision-diversity trade-off. Experiments on LEAN theorem proving demonstrate state-of-the-art results along the coverage-precision Pareto frontier.

**Strengths:**

- Clear conceptual contribution: The paper provides a unified perspective that clarifies how RLVR methods implicitly optimize Reverse KL to a filtered distribution, explaining diversity loss in a principled way.

- Theoretically grounded: The connection between RLVR and distributional matching (Lemmas 1-2) is well-established, and the use of α-divergences to interpolate between objectives is mathematically sound.

- Comprehensive experimental analysis: The diversity analysis using Simpson index and Shannon entropy, along with problem difficulty transitions, provides valuable insights into model behavior.

**Weaknesses:**

## Limited Novelty

The core technical contribution is incremental. The paper essentially applies existing f-DPG methods with Amari α-divergences to the RLVR setting. While the unification perspective is useful, the algorithmic novelty is limited:

- The connection between RLVR and Reverse KL was already established by Korbak et al. (2022b)
- f-DPG and KL-DPG are existing methods
- Amari α-divergences are well-known in the literature

The main contribution is demonstrating that α ≈ 1 can outperform standard GRPO, but this feels more like careful hyperparameter tuning than a fundamental advance.

Aligning Language Models with Preferences through f -divergence Minimization.

On Reinforcement Learning and Distribution Matching for Fine-Tuning Language Models with no Catastrophic Forgetting.

A Distributional Approach to Controlled Text Generation.


## Narrow Experimental Validation

- Single domain: Experiments are limited to LEAN theorem proving. The generalizability to other reasoning tasks (code generation, mathematical problem solving, multi-step reasoning) is unclear.

- Single base model: Only DeepSeek-Prover-V1.5-SFT (7B) is evaluated. Scaling behavior and performance on larger models is unknown.

## Overclaimed Improvements

- It missed a lot of relevant baselines: such as DAPO, Dr. GRPO, GPG, etc.

- Looking at Figure 2, the comparison is confusing. It seemed that Amari-DPG improved pass@1 but significantly harmed the pass@k when k becomes large (e.g., 256, 512). This can be understood as the model loses a lot of answer diversity.


## Overlook of Relevant Papers

The authors claim in the conclusion section that RLVR does not generate fundamentally new capabilities but instead reweights and amplifies behaviors already present in the base model. However, they failed to cite the very closely related papers that discovered the same phenomenon.

The invisible leash: Why rlvr may not escape its origin.

**Questions:**

How does the method perform on other reasoning domains beyond LEAN?

What is the computational overhead compared to GRPO in wall-clock time?

Can you provide statistical significance tests for the main results?

Why not compare to other recent LEAN training methods (e.g., Kimina-prover, InternLM2.5-StepProver)?

How sensitive are results to the ε threshold for Z_c?

---

> ### Author Response · Authors · 2025-11-25
>
> We thank the reviewer for the constructive feedback that has really helped improve our submission. We would like now address the points on which the reviewer noted this paper could be improved:
>
> ### Weaknesses
>
> > ** Limited Novelty**
>
> > The core technical contribution is incremental. The paper essentially applies existing f-DPG methods with Amari α-divergences to the RLVR setting. While the unification perspective is useful, the algorithmic novelty is limited
>
> We concur with the reviewer that our technical contribution is incremental, as we strongly build in prior art. We also note that this point was also shared with reviewer m7ni, who nonetheless weighed differently the value of this perceived weakness. In this respect, we would like to bring two more related points into consideration that add further nuance to the weight of this weakness.
> First, a paper can be incremental and yet hugely impactful. GRPO could be seen as an incremental modification of PPO, which was, in turn, an incremental modification of TRPO. Similarly, the only algorithmic innovation of the Korbak et al. (2022b) paper, was the introduction of the baseline to distributional policy gradients, but conceptually brought about the perspective on which we build here. Thus, incrementality is not per-se a good indicator of impact. Moreover, we are, to the best of our knowledge, the first to bring $\alpha$ divergences – which are not mainstream in ML – to the space of LLM post-training. The fact that we reuse prior art to derive the gradient estimator to minimize this divergence could be seen as good scientific practice as we make clear how our work fits within the existing body of knowledge.
> Even more importantly, we believe our paper should also be measured by its contribution to the conversation in the field. While these works have existed since 2023 and are relatively well-cited, to the best of our knowledge, there is **no comparison** to these techniques in any recent paper. Instead, all current work on training models using verifiable rewards focus on RL-only methods. Just by highlighting that distributional techniques are competitive or better than RL-based ones, this paper can greatly impact the research focus in the field.
>
> > The main contribution is demonstrating that α ≈ 1 can outperform standard GRPO, but this feels more like careful hyperparameter tuning than a fundamental advance.
>
> We regret that the reviewer got this impression from the paper. We don’t think this is a contribution of our paper, and that the main contributions lie elsewhere. In summary, beyond the conceptual contributions, which as we argued should not be neglected, we show that our technique finds the best available solutions when considering the tension between maximizing precision and preserving diversity. The model trained with α ≈ 1 achieves the best results in terms of maximizing pass@1, at the cost of pass@256. Yet, pass@256 is highly relevant for theorem proving: Fermat’s last theorem is not any less true by the fact that many mathematicians have failed at proving it. What matters is that there exists at least one correct proof. In that spirit, pass@256 tells us how many problems we can solve if we allow 256 attempts. Models trained with our technique are the best in this respect, surpassing highly competitive baselines such as pass@k training. We have rephrased the corresponding sentence in the enumeration of contributions to make our findings clearer.

---

> ### Author Response · Authors · 2025-11-25
>
> > **Narrow Experimental Validation**
>
> >Single domain: Experiments are limited to LEAN theorem proving. The generalizability to other reasoning tasks (code generation, mathematical problem solving, multi-step reasoning) is unclear.
>
> > Single base model: Only DeepSeek-Prover-V1.5-SFT (7B) is evaluated. Scaling behavior and performance on larger models is unknown.
>
> > How does the method perform on other reasoning domains beyond LEAN?
>
> Thank you for the suggestion to expand our empirical validation. We have now added in Appendix G new results for a Qwen-2.5-Math-1.5B trained on a subsection of the MATH dataset, and evaluated on the Minerva dataset, thus experimenting with a different model family of a different scale and a different domain.
>
> First of all, we still see from the results on pass@256 a diversity reduction effect but smaller for GRPO, which now has about the same coverage as the base model. Notably, Rewarding the Unlikely also shows a small effect, being on par with RLOO. $\alpha=0.999$, consistently with the previous experiment, performs similarly to these models. On the other hand, $\alpha=0.9$ achieves the highest pass@256, outperforming all other models, including the Pass@k-Training baseline. Lower values of alpha do not perform better in pass@256, which is puzzling for us. We conjecture that this comes from the fact that we did not pre-compute the partition function but rather computed it online on the basis of just 4 samples. We will confirm whether results change after precomputing the partition function by adding an ablation experiment for the final version. Note that noise on the partition function should not affect higher values of alpha (see the analysis on Appendix H).
>
> > **Overclaimed Improvements**
>
> > It missed a lot of relevant baselines: such as DAPO, Dr. GRPO, GPG, etc.
>
> Our GRPO baseline is in fact Dr. GRPO, which removes unintended biases in the objective and consistently performs better [1,2]. We have clarified this further in the paper to avoid confusion. We also thank the reviewer for the suggestion of including other RL baselines. Because methods such as DAPO, GSPO and GPG don’t depart significantly from GRPO, varying mostly in the clipping strategy, we have focused on one of them, namely GPG, which claims the best performance, but also included RLOO [3]  which is comparable to ours in that it also uses a leave-one-out-baseline and ReMax [4] which uses as a baseline the greedy solution from the model. These two other baselines performed surprisingly well, making it to the Pareto frontier. Still, our observation that $\alpha$-DPG trains models at the Pareto frontier spanning its full range, with a trade-off tunable by the parameter $\alpha$ remains true.
>
> [1] Liu, Zichen, Changyu Chen, Wenjun Li, et al. “Understanding R1-Zero-Like Training: A Critical Perspective.” arXiv:2503.20783.
> [2] Mistral-AI, Abhinav Rastogi, Albert Q. Jiang, et al. “Magistral.” arXiv:2506.10910.
> [3] Ahmadian, Arash, Chris Cremer, Matthias Gallé, et al. “Back to Basics: Revisiting REINFORCE Style Optimization for Learning from Human Feedback in LLMs.” arXiv:2402.14740.
> [4] Li, Ziniu, Tian Xu, Yushun Zhang, et al. “ReMax: A Simple, Effective, and Efficient Reinforcement Learning Method for Aligning Large Language Models.” arXiv:2310.10505.
>
> > Looking at Figure 2, the comparison is confusing. It seemed that Amari-DPG improved pass@1 but significantly harmed the pass@k when k becomes large (e.g., 256, 512). This can be understood as the model loses a lot of answer diversity.
>
> The behavior of $\alpha$-DPG depends on the specific value of $\alpha$. Higher values of $\alpha$ behave more like GRPO, starting with high pass@1 and flattening out for higher values of k. In contrast, lower values of $\alpha$ start with relatively lower pass@1, but achieve higher pass@256. For example, $\alpha=0.5$ achieves the highest performance in terms of pass@256, improving on the base model across all values of $\alpha$.
> There is a natural tension between the model’s precision, i.e., its ability to produce a correct solution in a single attempt, and its coverage, namely, its ability to produce at least one correct solution across many samples. With $\alpha$-DPG, we are **not** claiming to improve both metrics simultaneously. Instead, we show that we can pick an optimal trade-off between the two.

---

> > ### Author Response · Authors · 2025-11-25
> >
> > > Overlook of Relevant Papers: The authors claim in the conclusion section that RLVR does not generate fundamentally new capabilities but instead reweights and amplifies behaviors already present in the base model. However, they failed to cite the very closely related papers that discovered the same phenomenon.
> >
> > We do not claim to have discovered this phenomenon having cited the references that did [1,2,3]. Rather, our work offers an explanation for it and a solution that embraces this property as part of a well-principled objective function. Still, we do thank the reviewer for pointing us to “The invisible leash” paper, which we were not aware of, and have included it in our citations now. Notably, this paper only reinforces the need for studies such as ours, which illuminate the underlying mechanisms behind what they call “the invisible leash”. Also, to avoid confusion we have rephrased the cited sentence of the conclusions to make our contributions clearer.
> >
> > [1] Yue et al. 2025. Does Reinforcement Learning Really Incentivize Reasoning Capacity in LLMs Beyond the Base Model?
> > [2] He et al. 2025. Rewarding the Unlikely: Lifting GRPO Beyond Distribution Sharpening.
> > [3] Dang et al. 2025. Assessing Diversity Collapse in Reasoning
> >
> > ### Questions
> >
> > > What is the computational overhead compared to GRPO in wall-clock time?
> >
> > $\alpha$-DPG and GRPO are quite comparable in terms of wall clock time if not considering the computation of the partition function. However, the precomputation of the partition function for $\alpha$-DPG can induce considerable overhead as it is evaluated on fresh samples (in this case, about 10x as many as for a single run), which is is especially heavy if it will be used just once (on our Lean experiments we amortized this cost by training multiple models with the same pre-computations). However, because we were not training the models at the same time, we could leverage a higher number of GPUs and CPUs for its computation to reduce the time. For training in MATH we computed the partition function online on the basis of the training samples without an extra overhead, but we conjecture this might have affected lower values of alpha. For the final version, we will provide ablations on the need for pre-computing the partition function for both datasets.
> >
> > > Can you provide statistical significance tests for the main results?
> >
> > Thank you for the suggestion of strengthening the rigor of our empirical validation with significance tests, which are rare to see in the current landscape of ML research. We have measured significance between models using a paired bootstrap test, and added the results to a table in Appendix G. Additional Experimental Results.
> >
> > > Why not compare to other recent LEAN training methods (e.g., Kimina-prover, InternLM2.5-StepProver)?
> >
> > We thank the reviewer for proposing additional models and data to test our method. After reviewing them, we found that InternLM2.5-Step-Prover is not fully suited to our experimental setting. Its base model is not trained to generate full reasoning or complete proofs; instead, it is trained to predict the next tactic given the current proof state. This makes it well suited for tactic-level tree search solutions, but less appropriate for our reasoning setup, which requires multiple generation turns and rewriting to obtain a full proof. As a result, it would be more challenging to train it within the same framework used for DeepSeek Prover.
> > However, we are currently working on the extension of our experiments to Kimina-Prover with Numina-Lean data, which is computationally more intensive than our earlier setup because of longer generations which also require longer verification times. We will provide those additional results for the final version of the paper.
> >
> >
> > > How sensitive are results to the ε threshold for $Z_c$?
> >
> > For high values of $\alpha$ we can confidently say that the method is completely insensitive to the $\epsilon$ threshold. This is because, as we detail in the paper, the pseudo-reward becomes a soft approximation of the binary verifier value, independently of what the exact target probability should be. For lower values of $\alpha$, the $\epsilon$ value starts to play a more important role as the $p/\pi$ ratio is less attenuated by the $1-\alpha$ exponent. However, more crucial than the value of $\epsilon$ is the probability of eventually obtaining low-probability samples from the policy that make this ratio $p/\pi$ spike and can destabilize the training. For this reason, we have added a clipping value at the pseudo-reward level that prevents such spikes from destabilizing the training.

---

> ### Comment · Reviewer_yHrP · 2025-11-25
>
> Thanks for this detailed response. I really value the kind explanations point-to-point. Some of my concerns are solved, or partially addressed. However, I am worried about the so-called "trade-off" between pass@1 and pass@k. That is, if \alpha = 0.999, \alpha-DPG achieves the best in pass@1 but performs even worse than the base in pass@256. While if we select \alpha = 0.5, then \alpha-DPG outperforms all others in pass@256 but performs worse than GRPO. This makes me wonder:
>
> (1) Can the authors illustrate the applicability of different \alpha in different situations? When should we select high \alpha and when should we choose low \alpha? Which one do you prefer for the daily-use case?
>
> (2) I am on the authors' side that RLVR may be regarded more as a reweighting strategy instead of expanding the new reasoning capabilities. However, this is at present still a debated topic in the community, and some other papers like ProRL claim the boundary expansion after RLVR. How do the authors respond to the latter perspective? Actually, I believe the curve of \alpha-DPG is a very good example to prove the trade-off between diversity and precision, which supports the view of Yue et al. and Wu et al. (the leash paper). But I am just worried that some researchers in this community may disagree with it.

---

> > ### Author Response · Authors · 2025-11-26
> >
> > Thank you for following up on the discussion.
> >
> > > (1) Can the authors illustrate the applicability of different \alpha in different situations? When should we select high \alpha and when should we choose low \alpha? Which one do you prefer for the daily-use case?
> >
> > This is a great question that connects with reviewer m7ni’s question about whether there is a principled choice for selecting the value of $\alpha$. We believe there is depth to this question we haven’t yet fully uncovered, but we conjecture that different values of $\alpha$ should optimize for different pass@k, so you can pick the value of $\alpha$ to match your sampling budget. That said, one can measure the performance of pass@k at different values of $\alpha$ and make decisions on that information.
> >
> > With a limited sampling budget (e.g. a code generation service that must meet the demands from thousands of users), one should use a high value of alpha to maximize the chances of finding a correct solution within the allocated budget. With an extensive sampling budget (e.g., a mathematician using assisted theorem proving to solve hard problems), one should use a low value of alpha to ensure that one tests a diverse array of solutions in the hope of finding at least one that is correct.
> >
> >
> > > (2) I am on the authors' side that RLVR may be regarded more as a reweighting strategy instead of expanding the new reasoning capabilities. However, this is at present still a debated topic in the community, and some other papers like ProRL claim the boundary expansion after RLVR. How do the authors respond to the latter perspective? Actually, I believe the curve of \alpha-DPG is a very good example to prove the trade-off between diversity and precision, which supports the view of Yue et al. and Wu et al. (the leash paper). But I am just worried that some researchers in this community may disagree with it.
> >
> > This is science and we are all free to disagree with other researchers, providing reasoned arguments. Thank you also for sharing this other reference, which we also missed.
> >
> >
> >
> > The ProRL paper claims to generate novel solutions by 1) using a separate higher clipping threshold for the upper bound, and 2) resetting the reference policy every $n$ steps. Regarding point 1) our analysis is independent of the clipping strategy, which is done just for the purpose of keeping the training stable. Regarding point 2), let’s consider two extreme points: one in which $n=1$, so we update the policy on every step, and another in which we update the policy after a very large number of steps. In the first case, we can see we are in a case close to a PPO approach, where the reward function is defined purely as the verifier reward and the role of PPO clipping strategy, which regularizes $\pi_\theta$ to stay close to $\pi_{t-1}$,  is now taken by the regularization term $KL(\pi_\theta, \pi_{t-1})$ in the ProRL reward function. Now, let’s consider the latter case in which we allow a very large number $n$ of steps before resetting the policy. Then, we can suppose that the model initialized with $\pi_0$ has converged to the optimal distribution (see Lemma 1 in page 4 of our submission) $\pi_1(y) \propto \pi_0 \exp(v(y)/\beta)$. Then, we do this again, obtaining the distribution $\pi_2(y) \propto \pi_1 \exp(v(y)/\beta) = \pi_0 \exp(2 v(y)/\beta)$. In general, $\pi_k(y) \propto \pi_{k-1} \exp(v(y)/\beta) = \pi_0 \exp(k v(y)/\beta)$. Thus, we can see that we are making the target distribution more focused on the reward over time by annealing the $\beta$ parameter as $\beta/k$ where $k$ is the number of resets. This suggests a simpler alternative, which is to keep the reference model fixed and simply anneal the $\beta$ parameter towards 0 instead. Nonetheless, in both cases, we see we are not far from pure GRPO without KL regularization, so we find it unlikely that the proposed scheme ends up assigning much probability mass to solutions that were not already likely in the base model. A perplexity analysis such as the one we reproduced from [1], suggested by reviewer nArH, would be valuable in this respect.
> >
> > [1] Yue et al. 2025. Does Reinforcement Learning Really Incentivize Reasoning Capacity in LLMs Beyond the Base Model?

---

> > > ### Comment · Reviewer_yHrP · 2025-11-27
> > >
> > > Thanks for your response. I appreciate your efforts in addressing my concerns. I would like to lower my confidence and leave the decision to other reviewers' opinions.

---

### Official Review · Reviewer_bJDc · 2025-10-29

**Soundness:** 3
**Presentation:** 3
**Contribution:** 3
**Rating:** 6
**Confidence:** 4

**Summary:**

The paper presents DMVR, a distributional matching view for post training with verifiable rewards. The key idea is to form an explicit target distribution by filtering out incorrect generations with a verifier and then approximate this target with an autoregressive policy. The authors show that standard RL with verifiable rewards is equivalent to minimizing the reverse KL to a softened filtered target, which explains the loss of diversity in many RL runs. They then propose Amari DPG, which minimizes an α divergence to the filtered target and gives direct control over the precision and diversity tradeoff. Following theorem proving, the method traces a Pareto frontier between pass@1 and pass@256 and improves coverage over several GRPO style baselines.

**Strengths:**

1. The paper formalizes the filtered target distribution, pc, and proves that maximizing the usual RLVR objective equals minimizing the reverse KL to a softened filter. This clarifies why RLVR becomes mode seeking and why diversity collapses in practice. The argument is simple but convincing.
2. The method unifies rejection sampling fine-tuning at one end and RLVR-style training at the other end, and allows smooth interpolation between mass covering and mode-seeking regimes. This aligns with prior distributional-matching insights and provides a knob practitioners can use.
3. The paper also measures tactic and premise diversity and shows how those relate to pass@1 and pass@256, which is informative for understanding diversity effects beyond one score.

**Weaknesses:**

1. Evaluation is quite narrow. All main experiments are with a single base model family and a 7B scale, which is my biggest concern.
2. The GRPO baselines use β=0 by default and add a single high KL setting. However, recent works show that reward shaping like rewarding the unlikely or optimizing pass@k can rescue coverage. A stronger baseline suite with these variants tuned as carefully as Amari DPG would make the frontier result harder to question. [1] Rewarding the Unlikely: Lifting GRPO Beyond Distribution Sharpening

**Questions:**

1. How does Amari DPG compare against two recent diversity aware baselines: pass@k training and rewarding the unlikely, when all methods receive similar compute and careful tuning. If combined, do they add or conflict.
2. Several recent papers suggest that RL tends to reweight existing solutions rather than create new ones. Do you see evidence that α near zero brings the model closer to the base distribution in that sense, for example by recovering harder rare solutions that the base already had. Any analysis of overlap of successful trajectories would be helpful.
3. If possible, conduct more experiments on different model families. (I know it may be too much for a rebuttal, and I will not take this into my final rating, just for the paper quality considerations.)

---

> ### Author Response · Authors · 2025-11-25
>
> We thank the reviewer for the constructive feedback that has really helped improve our submission. We would like now address the points on which the reviewer noted this paper could be improved:
>
> ### Weaknesses
>
> > Evaluation is quite narrow. All main experiments are with a single base model family and a 7B scale, which is my biggest concern.
>
> > If possible, conduct more experiments on different model families. (I know it may be too much for a rebuttal, and I will not take this into my final rating, just for the paper quality considerations.)
>
> Thank you for the encouragement to expand our experimental validation. We have now added in Appendix G new results for a Qwen-2.5-Math-1.5B trained on a subsection of the MATH dataset, and evaluated on the Minerva dataset. First of all, we still see from the results on pass@256 a diversity reduction effect but smaller for GRPO, which now has about the same coverage as the base model. Notably, Rewarding the Unlikely also shows a small effect, being on par with RLOO. $\alpha=0.999$, consistently with the previous experiment, performs similarly to these models. On the other hand, $\alpha=0.9$ achieves the highest pass@256, outperforming all other models, including the Pass@k-Training baseline. Lower values of alpha do not perform better in pass@256, which is puzzling for us. We conjecture that this comes from the fact that we did not pre-compute the partition function but rather computed it online on the basis of just 4 samples. We will confirm whether results change after precomputing the partition function by adding an ablation experiment for the final version. Note that noise on the partition function should not affect higher values of alpha (see the analysis on Appendix H).
>
> > The GRPO baselines use β=0 by default and add a single high KL setting. However, recent works show that reward shaping like rewarding the unlikely or optimizing pass@k can rescue coverage. A stronger baseline suite with these variants tuned as carefully as Amari DPG would make the frontier result harder to question. [1] Rewarding the Unlikely: Lifting GRPO Beyond Distribution Sharpening
>
> > How does Amari DPG compare against two recent diversity aware baselines: pass@k training and rewarding the unlikely, when all methods receive similar compute and careful tuning. If combined, do they add or conflict.
>
> Pass@k training and rewarding the unlikely are indeed two of our main baselines. Please, refer to Figure 1, Figure 2, and Section 4 for details. We followed the same hyperparameter settings advocated by the original authors and reproduced similar results to them, thus we are comparing against strong baselines that considerably improve with respect to vanilla GRPO. Our method, however, further improves on top of them. We hope that this addresses the reviewer's concern, but in case it doesn’t, could the reviewer clarify it?
>
> ### Questions
>
> > Several recent papers suggest that RL tends to reweight existing solutions rather than create new ones. Do you see evidence that α near zero brings the model closer to the base distribution in that sense, for example by recovering harder rare solutions that the base already had. Any analysis of overlap of successful trajectories would be helpful.
>
> We agree, thank you for the suggestion! We have added to the submission a “Perplexity Analysis” paragraph in which we follow the setup of [1], measuring the perplexity according to the base model of RL/DPG tuned models. Our findings are in line with their finding that the sampled solutions remain likely under the base model. Furthermore, our analysis makes clear why this is the case, as the target distribution is the result of applying a filter on the base distribution and renormalizing.
>
> [1] Yue et al. 2025. Does Reinforcement Learning Really Incentivize Reasoning Capacity in LLMs Beyond the Base Model?

---

### Official Review · Reviewer_nArH · 2025-11-01

**Soundness:** 3
**Presentation:** 3
**Contribution:** 3
**Rating:** 4
**Confidence:** 3

**Summary:**

This paper addresses the diversity loss issue in LLMs tuned via Reinforcement Learning by proposing the Distributional Matching with Verifiable Rewards (DMVR)  framework. It constructs an explicit target distribution by filtering out incorrect answers while preserving the relative probabilities of correct ones from a pre-trained base model. To approximate this target, the paper introduces Amari-DPG, which leverages Amari’s α-divergence family to enable controllable trade-offs between precision and diversity.

**Strengths:**

1. The theoretical analysis is rigorous, clearly attributing the diversity loss in RL-tuned LLMs to the mode-seeking behavior induced by the Reverse KL divergence.

2. The paper is well-structured and addresses a meaningful research question, highlighting the significance of the accuracy–diversity trade-off in RL-aligned LLMs.

**Weaknesses:**

The main weakness lies in the experimental evaluation.

1. The authors focus on RL algorithms in LLMs, yet RL-specific experiments are limited. It is essential to include baselines covering algorithms such as PPO and REINFORCE.

2. Additionally, evaluations are confined to Lean. It would be valuable to incorporate 1–2 additional tasks (e.g., code generation or natural language mathematical reasoning) to assess the framework’s general applicability beyond formal theorem proving.

3. Why doesn't Lean's testset use mainstream theorem proving benchmark, such as MiniF2F？

If these concerns are adequately addressed, I will consider raising my score.

**Questions:**

1. Amari-DPG (α=0) uses on-policy sampling. Compared to RS-FT’s base model sampling, how does it perform in terms of sample efficiency (e.g., rejection rate of incorrect samples)?

2. Amari-DPG (α=0.25) shows limited improvement in the efficiency of medium-difficulty problems. Can dynamically adjusting α during training balance the goals of "preserving the solvability of hard problems" and "improving the efficiency of medium-difficulty problems"?

3. In Section 3.2, you argue that Amari’s α-divergence unifies Reverse KL  and Forward KL by interpolating between α→1 and α→0. However, the paper only shows gradient equivalence for these edge cases. Can you provide a formal proof (or more rigorous intuition) that the α-divergence’s interpolation smoothly preserves the "mode-seeking to mass-covering" transition across all $\alpha \in (0,1)$ ?

4. You unify RLVR and RS-FT by showing Amari-DPG recovers both methods at α extremes. However, RLVR uses a KL penalty ($\beta$) to balance reward and proximity to $\pi_{base}$, while Amari-DPG sets $\beta=0$ by default. Theoretically, how does Amari-DPG’s α parameter interact with RLVR’s $\beta$ if both are used together?

---

> ### Author Response · Authors · 2025-11-25
>
> We thank the reviewer for the constructive feedback that has really helped improve our submission. We would like now address the points on which the reviewer noted this paper could be improved:
>
> ### Weaknesses
>
> > The authors focus on RL algorithms in LLMs, yet RL-specific experiments are limited. It is essential to include baselines covering algorithms such as PPO and REINFORCE.
>
> Thank you for encouraging us to include more RL-specific baselines. The difference between REINFORCE, PPO and GRPO typically lies in the baseline that is used to reduce the variance of the gradient estimate. As it is commonly understood, REINFORCE does not foresee the usage of any baseline, inducing estimates with larger variance and thus, is less stable and a poor baseline. PPO, on the other hand, requires training a value network which is used as a baseline, which makes it more complex and expensive to train. This is why all recent work focuses on critic-free methods such as GRPO, which uses a group average of the rewards as a baseline. In the revised version, we have clarified that we also focus on such critic-free methods.
> To address the reviewer’s concerns while also focusing on stronger contenders, we have decided to include three other RL-specific training techniques: RLOO [1], ReMax [2] and GPG [3]. The first one is interesting because it uses an unbiased estimator of the RL loss (as opposed to the biased GRPO). The second one uses as a baseline the reward from a greedily-generated response. The third one is a minor variation of GRPO that is claimed to improve performance. Interestingly, RLOO and ReMax performed very well. RLOO is almost on par with our $\alpha=0.999$ model, whereas ReMax also makes it to the Pareto frontier. GPG, on the other hand, performed worse than GRPO. Our observation that $\alpha$-DPG produces models at the Pareto frontier remains true for alpha values above a given threshold and, in contrast to RL baselines, span its full range. We refer the reviewer to the revised submission for further details and we hope that the inclusion of these baselines addresses their concerns.
>
> [1] Ahmadian, Arash, Chris Cremer, Matthias Gallé, et al. “Back to Basics: Revisiting REINFORCE Style Optimization for Learning from Human Feedback in LLMs.” arXiv:2402.14740.
> [2] Li, Ziniu, Tian Xu, Yushun Zhang, et al. “ReMax: A Simple, Effective, and Efficient Reinforcement Learning Method for Aligning Large Language Models.” arXiv:2310.10505.
> [3] Chu, Xiangxiang, Hailang Huang, Xiao Zhang, Fei Wei, and Yong Wang. “GPG: A Simple and Strong Reinforcement Learning Baseline for Model Reasoning.” arXiv:2504.02546.
>
> > Additionally, evaluations are confined to Lean. It would be valuable to incorporate 1–2 additional tasks (e.g., code generation or natural language mathematical reasoning) to assess the framework’s general applicability beyond formal theorem proving.
>
> Thank you for the encouragement to expand our experimental validation. We agree both code generation and informal mathematical reasoning can both benefit from our approach. Because code generation seems to require large computation budgets (large GPU clusters to process 16k+ tokens outputs and a large number of CPUs for processing the test cases), we have decided to focus on the latter.
> We have now added in Appendix G new results for a Qwen-2.5-Math-1.5B trained on a subsection of the MATH dataset, and evaluated on the Minerva dataset. First of all, we still see from the results on pass@256 a diversity reduction effect but smaller for GRPO, which now has about the same coverage as the base model. Notably, Rewarding the Unlikely also shows a small effect, being on par with RLOO. $\alpha=0.999$, consistently with the previous experiment, performs similarly to these models. On the other hand, $\alpha=0.9$ achieves the highest pass@256, outperforming all other models, including the Pass@k-Training baseline. Lower values of alpha do not perform better in pass@256, which is puzzling for us. We conjecture that this comes from the fact that we did not pre-compute the partition function but rather computed it online on the basis of just 4 samples. We will confirm whether results change after precomputing the partition function by adding an ablation experiment for the final version. Note that noise on the partition function should not affect higher values of alpha (see the analysis on Appendix H).

---

> > ### Author Response · Authors · 2025-11-25
> >
> > > Why doesn't Lean's testset use mainstream theorem proving benchmark, such as MiniF2F？
> >
> > We agree that MiniF2F is a standard benchmark for Lean theorem proving. As also reported by authors from [Rewarding the unlikely](https://arxiv.org/pdf/2506.02355),  MiniF2F introduces high variance, strong distribution shift, and large difficulty gaps, especially at our training scale. Small changes in training runs produce inconsistent MiniF2F results, making it difficult to draw clear conclusions about the effect of our algorithm itself.
> > To ensure stable, interpretable, and comparable results, we reproduced the exact same experimental setup as the Rewarding the unlikely paper, conserving the same train and val set (IID held-out set), exactly as done in the original work. MiniF2F is better suited for final large-scale benchmarks rather than controlled studies like ours.
> >
> >
> > ### Questions
> >
> > > Amari-DPG (α=0) uses on-policy sampling. Compared to RS-FT’s base model sampling, how does it perform in terms of sample efficiency (e.g., rejection rate of incorrect samples)?
> >
> > Thank you for this interesting question. We computed this by summing the average reward at each step for DPG and compared it to the pass@1 estimate of the base model (0.5423) multiplied by the total number of steps (200), as an estimate of the RS-FT sample efficiency. This results in an 8% increase in sample efficiency for DPG. We note that this number highly depends on the pass@1 of the base model. Here, the base model is relatively proficient at producing correct responses, which is why the relative improvement is modest. However, for models that rarely produce the correct output, the relative improve
> >
> > > Amari-DPG (α=0.25) shows limited improvement in the efficiency of medium-difficulty problems. Can dynamically adjusting α during training balance the goals of "preserving the solvability of hard problems" and "improving the efficiency of medium-difficulty problems"?
> >
> > Lower values of $\alpha$ indeed have a smaller effect at increasing precision, reflected in the amount of medium problems becoming easy or of hard problems becoming medium or easy. The counterpart of that is that such values retain coverage and even increase it. This is reflected in the hard problems that remain solvable and unsolvable problems that do become solvable. Higher values of $\alpha$ have the opposite effect.
> >
> > We agree it would be interesting to test whether by slowly increasing alpha, in a similar spirit to curriculum-learning, we can improve precision while retaining higher levels of coverage (incidentally, reviewer m7ni had a similar thought). However, this doesn’t necessarily need to be the case. The choice of divergence impacts two different aspects: the optimization dynamics and the optimal solution within a restricted parametric family. For a parametric family that includes $p$ and a sufficiently regular optimization landscape, the choice of divergence shouldn’t matter as the policies should all converge to $p$. However, for a rough landscape and a parametric family that doesn’t include $p$, we can ask whether the resulting model’s properties are a consequence of the training dynamics or of the model that minimizes the given divergence. If by dynamically adjusting alpha we end up with very different models this would hint that training dynamics matter, and otherwise, that it is the optimal model in the parametric family that dominates. Either way, it would be very interesting to find out! Answering this question in depth, however, would require considerable empirical work that goes beyond the scope of this work and so, we leave it as an interesting direction to explore in future work. Thank you for the suggestion!

---

> > > ### Author Response · Authors · 2025-11-25
> > >
> > > > In Section 3.2, you argue that Amari’s α-divergence unifies Reverse KL and Forward KL by interpolating between α→1 and α→0. However, the paper only shows gradient equivalence for these edge cases. Can you provide a formal proof (or more rigorous intuition) that the α-divergence’s interpolation smoothly preserves the "mode-seeking to mass-covering" transition across all $\alpha \in (0,1) $ ?
> > >
> > > We thank you for this important question, which encouraged us to dig deeper into the formal properties of our setup, and to add section H (Formal complements on $\alpha$-divergence) in our Appendix.
> > > The fact that $D_{f_\alpha}(\pi,p)$ is a continuous function of $\alpha$ and that it converges to the forward $KL(p||\pi)$ for $\alpha \to 0$ and to the reverse $KL(\pi||p)$ for $\alpha \to 1$ is well-known in the literature, e.g. [Cichocki and Amari (2010)].
> > >
> > > In our specific situation, while typically the autoregressive policy $\pi$ is full-support ($\pi(y)>0, \forall y\in \mathcal{Y}$), the support $A := \{y : p(y) > 0\}$ of $p$ is a proper subset of $\mathcal{Y}$. In such cases, while $KL(p||\pi)$ is finite, $KL(\pi||p)$ is infinite.
> > >
> > > In order to understand how different values of $\alpha$ select for different policies, it is instructive to compare $D_{f_\alpha}(\pi,p)$ with $D_{f_\alpha}(\pi',p)$ for two candidate policies $\pi$ and $\pi'$.
> > >
> > > We provide an illustration (see section H of the Appendix) on a toy example with $\mathcal{Y} = \{y_1,y_2,y_3\}$ and $A = \{y_1, y_3\}$, and with three policies. The policy $\pi_1$ is more “covering” than $\pi_2$ and $\pi_3$, with a lower forward $KL(p||\pi)$, but it is less “focussed” on the valid region $A$ than either $\pi_2$ or $\pi_3$, which are both concentrated on $A$, but with different peaks. Despite the fact that the reverse $KL(\pi||p)$ is infinite for the three policies, for $\alpha$ close to $1$ the divergences, while large (and tending to infinity), still show a clear order, with $D_{f_\alpha}(\pi_1,p)$ much higher than $D_{f_\alpha}(\pi_3,p)$, which in turn is slightly higher than $D_{f_\alpha}(\pi_2,p)$. For lower values of $\alpha$, the “mass-covering” $\pi_1$ is preferred to the other two.
> > >
> > > The same Section H contains a formal (and apparently novel) result, which permits a better understanding of what happens in the situation where the support $A$ of the target $p$ is contained in the support of the model $\pi$. This result says that, with $\pi(A)$ the $\pi$ mass of $A$, and with $\pi_A$ is the “renormalization” of $\pi$ to $A$, that is, $\pi_A(y) = \pi(y)/\pi(A)$ for $y\in A$, and for $\alpha\in (0,1)$, we have the identity
> > >
> > > $D_{f_\alpha}(\pi, p) = \frac{1 - \pi(A)^\alpha}{\alpha(1 - \alpha)} + \pi(A)^\alpha D_{f_\alpha}(\pi_A, p)$.
> > >
> > > This identity is especially interesting for the case of $\alpha$ tending to $1$. In that case, with $\pi$ full support and $\pi(A) < 1$, the support of $\pi_A$ is equal to the support of $p$, and therefore $D_{f_\alpha}(\pi_A, p)$ tends to a finite value $KL(\pi_A,p)$, and the second term of the identity tends towards a finite value. On the other hand, the first term tends to infinity at a rate close to $\frac{1-\pi(A)}{1-\alpha}$, meaning that $\pi(A) > \pi’(A)$ implies that the divergence of $\pi$ is lower than the divergence of $\pi’$ in the limit. In other words, if $A$ is the subset of $\mathcal{Y}$ for which the binary reward $v(y)$ is equal to $1$, and for $\alpha$ sufficiently close to $1$, minimizing $D_{f_\alpha}(\pi_\theta, p)$ is essentially equivalent to maximizing $\mathbb{E}{\pi\theta} v(y)$, the same objective as pure REINFORCE. On the other hand, as $\alpha$ tends to 0, the first term tends to $-\log(\pi(A))$ while the second tends to KL(p||\pi_A), which strongly penalizes $y$’s s.t. $\pi_A(y) \ll p(y)$, which penalizes lack of coverage of $p$. The equation makes clear the smooth interpolation between “support leakage penalization” on one extreme ($\alpha \rightarrow 1$) and “lack of coverage penalization” on the other ($\alpha \rightarrow 0$).

---

> > > > ### Author Response · Authors · 2025-11-25
> > > >
> > > > > You unify RLVR and RS-FT by showing Amari-DPG recovers both methods at α extremes. However, RLVR uses a KL penalty ($\beta$) to balance reward and proximity to $\pi_{base}$, while Amari-DPG sets $\beta=0$ by default. Theoretically, how does Amari-DPG’s α parameter interact with RLVR’s $\beta$ if both are used together?
> > > >
> > > > This is a very interesting question! First, note that these two parameters modulate different aspects of the loss landscape. $\alpha$ chooses among a family of different divergences that optimize the model towards a fixed target distribution. $\beta$, on the other hand, shapes the target distribution itself. With higher values of $\beta$ the target distribution is closer to the base distribution, making it easier to approximate with an autoregressive policy that was initialized with the base model, so that we speculate that the choice of $\alpha$ has smaller impact.
> > > >
> > > > We note, however, that a distribution with higher parameter $\beta$ is not a good description of our real objective, as it also assigns probability mass to invalid solutions to the target problem.

---

### Official Review · Reviewer_m7ni · 2025-11-01

**Soundness:** 3
**Presentation:** 3
**Contribution:** 3
**Rating:** 8
**Confidence:** 3

**Summary:**

Building on prior works on Distributional Matching, the authors consider the target distribution $p_c$, which is essentially the I-projection of the base policy onto the space of distributions that strictly filters out incorrect responses. The authors then argue that the RLVR optimizes the reverse KL to $p_c$, which potentially reduces diversity. In order to interpolate between the reverse and forward KL behaviors, the authors propose **Amari-DPG**, which utilizes Amari's $\alpha$-divergence for the $f$-DPG (Go et al., 2023). The efficacy of this is shown in a formal math dataset with Lean, along with extensive ablations.

**Strengths:**

- Paper is well-written and easy to follow, with enough background explanation
- A simple approach with meaningful peformance gain
- Extensive ablations

**Weaknesses:**

- Novelty-wise, at the end of the day, the methodology is precisely f-DPG of Go et al. (2023), with f-divergence replaced with Amari's $\alpha$-divergence. Also, the target distribution $p_c$ is taken from Khalifa et al. (2021). But, I also concur that the simplicity of the proposed method overshadows the "lack" of novelty.
- The writing can be made a bit clearer. As this is a mixture of prior works, a clearer separation of the authors' contributions and prior works in Section 3 would be helpful.
- The computation of the partition function, $Z_c$, is not mentioned explicitly anywhere in the paper. This makes the paper not so self-contained.

**minor suggestion**
- The "original" reference for the alpha-divergence is [1], and thus also referred to as Renyi's alpha-divergence. But, their forms are a bit different, although they are equivalent up to reparametrization; see Appendix B of [2], for instance. Maybe good to mention this just for clarity..!


[1] https://projecteuclid.org/ebooks/berkeley-symposium-on-mathematical-statistics-and-probability/On-Measures-of-Entropy-and-Information/chapter/On-Measures-of-Entropy-and-Information/bsmsp/1200512181

[2] https://proceedings.mlr.press/v70/li17a.html

**Questions:**

1. What may be a principled way of choosing $\alpha$ other than doing a grid search of $\alpha$?

2. Does different value of $\alpha$ impact the trainig stability/convergence rate?

3. Albeit with extensive ablations, the authors only consider the formal math task. What other tasks may benefit significantly from the proposed approach?

4. Can one think about "tuning" the $\alpha$ throughout the training? Of course, this means that the partition function would have to be reevaluated, but just a random thought.

5. (Minor) Very recently, there have been some issues regarding floating-point operations (https://x.com/QPHutu/status/1984258808332550245). Can the authors comment on whether this issue(?) applies to their experiments?

6. (Minor) Other than Amari's $\alpha$-divergence, there is another way of interpolating forward and reverse KL via density ratio metrics (DRMs) [1]. Could the authors comment on what would happen if such another interpolation is used?

---

> ### Author Response · Authors · 2025-11-25
>
> We thank the reviewer for their thoughtful feedback which has really helped improve our submission, and for the overall positive evaluation of our work. We would like now address the points on which the reviewer noted this paper could be improved:
>
> > Novelty-wise, at the end of the day, the methodology is precisely f-DPG of Go et al. (2023), with f-divergence replaced with Amari's $\alpha$-divergence. Also, the target distribution is taken from Khalifa et al. (2021). But, I also concur that the simplicity of the proposed method overshadows the "lack" of novelty
>
> We concur with the reviewer that our technical contribution is incremental, as we strongly build in prior art. We also thank the reviewer for hedging the strength of the point by noting (positively!) the simplicity of the approach. However, because reviewer yHrP had a similar observation albeit weighing it differently, we would like to bring forward two more related points into consideration that add further nuance to the weight of this weakness.
>
> First, a paper can be incremental and yet hugely impactful. GRPO could be seen as an incremental modification of PPO, which was, in turn, an incremental modification of TRPO. Similarly, the only algorithmic innovation of the Korbak et al. (2022b) paper, was the introduction of the baseline to distributional policy gradients, but conceptually brought about the perspective on which we build here. Thus, incrementality is not per-se a good indicator of impact. Moreover, we are, to the best of our knowledge, the first to bring $\alpha$ divergences – which are not mainstream in ML – to the space of LLM post-training. The fact that we reuse prior art to derive the gradient estimator to minimize this divergence could be seen as good scientific practice as we make clear how our work fits within the existing body of knowledge.
>
> Even more importantly, we believe our paper should also be measured by its contribution to the conversation in the field. While these works have existed for a few years and are relatively well-cited, to the best of our knowledge, there is no comparison to these techniques in any recent paper. Instead, all current work on training models using verifiable rewards focus on RL-only methods. Just by highlighting that distributional techniques are competitive or better than RL-based ones, this paper can greatly impact the research focus in the field.
>
> > clearer separation of the authors' contributions and prior works in Section 3
>
> We agree that this could be improved. We added the following sentence at the end of Section 3’s introduction to clarify these aspects: “Of these, points (1) and (3) are original to our work, whereas point (2) is reproduced from Korbak et al. (2022b). Moreover, the target distribution and the f -DPG technique to approximate it were defined in prior art (Khalifa et al., 2021; Go et al., 2023), as detailed in Section 2.” As a reminder, point (1) was the equivalence of the KL Control target distribution using parameter $\beta$ and the binary filter distribution, (2) The equivalence of RLVR and reverse KL optimization and (3) the usage of $\alpha$ divergences.
>
> We hope that these clarifications will make the separation of contributions clearer to the reader without breaking the flow of the argument.
>
> > The computation of the partition function, $Z_c$, is not mentioned explicitly anywhere in the paper. This makes the paper not so self-contained
>
> Thank you for pointing this out. We have included a description for how it can be computed in Appendix E, and referenced it in the paper at the points in which we mention the partition function.
>
> > Suggestion about mentioning Renyi divergence in relation to Amari
>
> Thank you for the suggestion, we have updated the submission to reflect this connection, as well as citing the Cressie Read divergence which is also equivalent up to reparametrization. Furthermore, we have renamed our method to $\alpha$-DPG to acknowledge the more general character of $\alpha$ divergences. We have also exploited in our new Appendix section H the connection between $\alpha$-divergences and the expression $\sum_y p(y)^\alpha q(y)^\alpha$, mentioned in Appendix B of [2], in order to prove a ``decomposition theorem” in response to reviewer NArH.

---

> > ### Author Response · Authors · 2025-11-25
> >
> > ### Questions
> >
> > > What may be a principled way of choosing $\alpha$ other than doing a grid search of $\alpha?
> >
> > This is a great question, thank you! We think there might be some monotonic relation between what we call coverage and precision, which would allow for using binary search to efficiently find an alpha that better approximates the desired trade-off. Another related question could be deciding which $\alpha$ would maximize pass@k for a given $k$. We would need to investigate these questions further to provide a definitive answer.
> >
> > > Does different value of $\alpha$ impact the training stability/convergence rate ?
> >
> > Yes, while for high values of alpha the pseudo-reward approximates a soft binary reward, as argued in Section 3 of the paper, for low values of alpha, the $p/\pi$ ratios in the pseudo reward can display large variance, especially during latter stages of training, destabilizing the rewards. This is in fact probably what happened with our $\alpha=0$ model in the original submission. We have updated the method and the experiments in the submission to include a clipping value of the pseudo-reward to improve the method’s stability.
> >
> > > Albeit with extensive ablations, the authors only consider the formal math task. What other tasks may benefit significantly from the proposed approach?
> >
> > Thank you for the encouragement to expand our experimental validation. We have now added in Appendix G new results for a Qwen-2.5-Math-1.5B trained on a subsection of the MATH dataset, and evaluated on the Minerva dataset. First of all, we still see from the results on pass@256 a diversity reduction effect but smaller for GRPO, which now has about the same coverage as the base model. Notably, Rewarding the Unlikely also shows a small effect, being on par with RLOO. $\alpha=0.999$, consistently with the previous experiment, performs similarly to these models. On the other hand, $\alpha=0.9$ achieves the highest pass@256, outperforming all other models, including the Pass@k-Training baseline. Lower values of alpha do not perform better in pass@256, which is puzzling for us. We conjecture that this comes from the fact that we did not pre-compute the partition function but rather computed it online on the basis of just 4 samples. We will confirm whether results change after precomputing the partition function by adding an ablation experiment for the final version. Note that noise on the partition function should not affect higher values of alpha (see the analysis on Appendix H).
> >
> > > Can one think about "tuning" the $\alpha$ throughout the training? Of course, this means that the partition function would have to be reevaluated, but just a random thought.
> >
> > We agree it would be interesting to test whether by slowly increasing alpha, in a spirit akin to curriculum learning, we can improve precision while retaining higher levels of coverage –notably, reviewer NArH had a similar thought–. However, this doesn’t necessarily need to be the case. The choice of divergence impacts two different aspects: the optimization dynamics and the optimal solution within a restricted parametric family. For a parametric family that includes $p$ and a sufficiently regular optimization landscape, the choice of divergence shouldn’t matter as they should all converge to $p$. However, for a rough landscape and a parametric family that doesn’t include $p$, we can ask whether the resulting model’s properties are a consequence of the training dynamics or of the model that minimizes the given divergence. If by dynamically adjusting alpha we end up with very different models this would hint that training dynamics matter, and otherwise, that it is the optimal model in the parametric family that dominates. Either way, it would be very interesting to find out! Answering this question in depth, however, would require considerable empirical work that goes beyond the scope of this work and so, we leave it as an interesting direction to explore in future work. Thank you for your suggestion!
> >
> > NB: The partition function wouldn’t need to be re-evaluated as $\alpha$ is a parameter of the divergence and not of the target distribution.

---

> > > ### Author Response · Authors · 2025-11-25
> > >
> > > > (Minor) Very recently, there have been some issues regarding floating-point operations (https://x.com/QPHutu/status/1984258808332550245). Can the authors comment on whether this issue(?) applies to their experiments?
> > >
> > > We enthusiastically thank the reviewer for pointing out this issue regarding the bfloat16! We were not aware of reported floating-point operation issues, and this could very-well be relevant for us as we heavily rely on importance sampling estimates that could be destabilized by floating point errors. While we did observe training instabilities in our experiments, especially for low values of $\alpha$ (now reported in the limitations paragraph), they seem to be related to variance of the pseudo-reward. We have quickly verified whether using fp16 single-handedly addresses those issues, but we couldn’t find evidence that it did. However, we are still investigating the implications of this issue, as we believe it could explain instabilities when sampling longer sequences. In the meantime, we have included a note on this in the limitations section.
> > >
> > > > (Minor) Other than Amari's $\alpha$-divergence, there is another way of interpolating forward and reverse KL via density ratio metrics (DRMs) [1]. Could the authors comment on what would happen if such another interpolation is used?
> > >
> > > We thank the reviewer for calling our attention to DRMs, which offer a larger perspective on defining divergences between distributions [3]. Beyond $f$-divergences, which do centrally exploit density ratios of the form $r(x)= p(x)/q(x)i$, some recent works in this tradition [4] provide an approach where certain IPMs (Integral Probability Metrics, which include the Wasserstein distance) can be seen as a midpoint in a range between the forward and the reverse KL. It would indeed be interesting to analyse the behavior of such DRMs in our context, and in particular their connection to moment-matching techniques, which are prominent in some works that inspired us, such as Khalifa et al (2021).
> > >
> > > [3] Sugiyama et al. (2010). Density Ratio Estimation: A Comprehensive Review.
> > >
> > > [4] Kato et al. (2023). "Unified Perspective on Probability Divergence via the Density-Ratio Likelihood: Bridging KL-Divergence and Integral Probability Metrics”

---

> > > > ### Comment · Reviewer_m7ni · 2025-11-27
> > > >
> > > > I thank the authors for the responses, which have addressed all of my concerns. Hence, I retain my original score.

---

### Author Response · Authors · 2025-11-25
**To all reviewers**

We thank the reviewers for their constructive feedback, which has really helped us to improve our submission on multiple aspects. We are encouraged that the reviewers found the "paper is well-written and easy to follow" (m7ni), "well-structured" (nArH), and "theoretically grounded" (yHrP). We are pleased that reviewers recognized the "clear conceptual contribution" (yHrP), noting that "the theoretical analysis is rigorous" (nArH) and "the argument is simple but convincing" (bJDc). Reviewers agreed that the paper "provides a unified perspective" (yHrP) that "clarifies why RLVR becomes mode seeking" (bJDc). Finally, we are glad reviewers appreciated the "extensive ablations" (m7ni) and "comprehensive experimental analysis" (yHrP)—noting the paper "measures tactic and premise diversity" (bJDc)—which "provides valuable insights" (yHrP) and is "informative for understanding diversity effects" (bJDc).

We now wish to make a summary of the updates we have made to the paper in response to their feedback:

- (Erratum) We noted that the samples generated for the evaluation in our original submission had non-neutral sampling parameters (temperature=0.9, top-k=50), which biased the pass@k estimations. Furthermore, a few checkpoints were trained for different numbers of steps because of a bug. We solved both of these issues in the updated submission and apologize for the mistake. Our new results are still mostly in line with the previously reported ones, with the only difference of $\alpha=0.5$ which now achieves the best pass@256 performance. Also GRPO with “rewarding the unlikely” is now at the Pareto frontier whereas it was only slightly farther away before.
- We introduced three new RL baselines (GPG, RLOO and ReMax) following the suggestions of reviewers bJDc and yHrP. Of these, RLOO and ReMax also make it to the Pareto frontier, albeit confined to the rightmost end (lower coverage). As before, there is a value of $\alpha$ above which all $\alpha$-DPG models remain Pareto-optimal or very close to the frontier and span its full range.
- We included new results on informal mathematical reasoning in Appendix G following the request from all reviewers.
- We included a new perplexity analysis paragraph following the suggestion of reviewer NArH.
- We included new theoretical results in Appendix H that characterize alpha divergences in our specific case in which the target distribution’s support is strictly included in that of the policy (following a question from nArH).
- We moved bootstrap variance estimates of the performances reported in Figure 1 to the appendix.
- We added statistical significance results (as suggested by yHrP).
- We added a known limitations section after the conclusions to make clear which aspects of our approach could be improved in the future.
- We renamed Amari-DPG to $\alpha$-DPG, which we believe is a clearer name noting that Amari $\alpha$ divergence is equivalent to Rényi (or Cressie-Read) $\alpha$ divergence up to reparametrization (as noted by m7ni).
- We renamed the context variable $c$ as $x$ to match existing literature.
- We described more in detail how the partition function $Z_x$ is computed in Appendix E.
- We have emphasized that we are in the space of critic-free techniques.

For convenience, the main changes have been highlighted in dark red.

---

### Author Response · Authors · 2025-12-03
**Summary for the Area Chair**

### Global summary
Our submission focuses on the issue that using **KL-regularized reinforcement learning** to maximize the average proportion of correct answers according to some verifier function (a metric that we denote by “precision”)  **leads to a marked reduction in the diversity of responses**. This reduction is particularly serious in applications such as theorem proving, as it damages the “coverage” (number of problems that can be solved). We leverage the observation that **KL-regularized RL is equivalent to minimizing the reverse KL** to a well-defined target distribution, and argue that the “mode-seeking” or “zero-forcing” nature of reverse KL is what causes the reduction of diversity. Next, to solve the problem of diversity collapse, we propose to **interpolate smoothly between the reverse KL and the “mass-covering” or “zero-avoiding” forward KL**, which naturally preserves more diversity, **using $\alpha$-divergences** for this purpose. The approach is denoted by $\alpha$-DPG and we validate it on a formal theorem proving task using Lean. We show that **$\alpha$-DPG finds pareto-optimal solutions, including those with the best overall coverage**.

Our paper was praised by the reviewers for bringing about **clear theoretically-founded conceptual insights** and offering a **simple unifying framework**, supported by a **comprehensive experimental analysis**. Furthermore, thanks to the reviewers feedback we have **improved the submission** on multiple axes. Notably, we have 1) **extended the experimental validation** of the approach **on mathematical informal reasoning** (training on the MATH dataset and evaluating on Minerva), and report preliminary results on a **new combination of model and data based on KiminaProver** for the formal reasoning experiments, 2) included **three new baselines** (GPG, RLOO and ReMax), 3) added **additional theoretical analysis in Appendix H**, and 4) added **additional empirical support** for our claims including a perplexity analysis of the generated responses and significance results. All in all, these additions have strongly improved our paper, while reinforcing our claims relative to the performance of $\alpha$-DPG.



**Initial scores and events during the discussion**:

- m7ni: 8 [The reviewer confirmed this score during the discussion session]
- bJDc: 6 [No response from the reviewer after the interruption]
- nArH: 4 [No response from the reviewer after the interruption]
- yHrP: 4 [The reviewer appreciated our responses; they lowered their confidence to 1, leaving the decision to the other reviewers]

**For additional high-level details, please look below at the overview of the paper, of the revision, and of the discussion before the interruption.**

---

> ### Author Response · Authors · 2025-12-03
>
> ### Our paper in a nutshell
>
> In this paper we focus on **training a model $\pi_{base}$ to generate solutions $y$ to a problem $x$ given the signal of a verifier $v(y,x) \in \\{0,1\\}$**. This task is **predominantly, if not exclusively, tackled using RL** by maximizing the expected verifier score plus an optional regularization term. Recent studies [4,5] have identified that this approach results in a **considerable loss of diversity** in the trained models. Here, we build on the earlier observation from [2] that KL-regularized RL is equivalent to minimizing the _Reverse KL_ divergence to the target distribution $p_{x,\beta}$ defined as
>
> $p_{x,\beta}(y) \propto \frac{1}{Z_x(\beta)} \pi_{base}(y\mid x)\exp\big(v(y,x)/\beta\big)$
>
> Then, we argue that the ideal target distribution is in fact simply the I-projection of $\pi_{base}$ into the manifold of distributions that meet $v(y,x)=1$ for all $y$ in their support, which was earlier proposed by [1] in the context of applying general constraints:
>
> $p_x(y) \propto \pi_{base}(y|x) v(y,x)$
>
> Furthermore, **we argue that the loss of diversity is a consequence of optimizing the “mode-seeking” or “zero-forcing” Reverse KL, and instead propose using $\alpha$ divergences, which smoothly interpolate between Reverse KL ($\alpha=1$) and the “mass-covering” or “zero-avoiding” Forward KL ($\alpha=0$) to conserve more of the base distribution diversity**. We implement this algorithm using the f-DPG algorithm from [3], calling it **Amari-DPG** in the original submission, later revised to **$\alpha$-DPG** after feedback from m7ni.
>
> We adopt theorem proving using the Lean environment as our experimental setup. For this, we follow [4], training DeepSeek-Prover-V1.5-SFT, a 7B model fine-tuned on theorem-proving data on 10k solvable Lean problems extracted from the Lean Workbook Dataset, and evaluate on the mean pass@k. We denote mean pass@1 “precision”, which is the probability that for any given problem, the model will produce a correct solution in a single attempt. On the other hand, following prior work [5] we consider mean pass@256 as a measure of coverage, which quantifies how many problems on average can be solved given a sufficiently large sample budget. Crucially, **$\alpha$-DPG can produce models that are at the Pareto frontier when trading off precision for coverage**, and **achieve maximum coverage for some values of $\alpha$**. We finally show, using quantitative metrics of diversity, correlations with pass@1/pass@256, which validate the connection between diversity and pass@k.
> ### Revision
> Following the reviewers feedback, we have improved the paper in the following main directions:
>
> - **Additional experiments**: We have included additional experiments both on training on the MATH dataset while evaluating on the Minerva dataset, and some preliminary results using models and data from Kimina-Prover [9]. The latter are computationally more intensive than our earlier experiments and so, we will complete the evaluation for the final version.
> - **Additional baselines**: We have added three additional baselines, namely GPG [10], RLOO [11] and ReMax [12]. Of these, RLOO and ReMax show pareto-optimal results together with $\alpha$-DPG models.
> - **Additional theoretical analysis**: We have included a new Appendix H that provides further theoretical insights on the behaviours of $\alpha$-divergences in our specific case in which the target distribution’s support is strictly included in the support of the model.
> - **Additional empirical analysis**: We have replicated the perplexity analysis of [5] for the Amari models, confirming that the samples generated by the trained models had high probability already in the base models, and that the spread is higher for lower values of $\alpha$. We have also included significance tests.

---

> > ### Author Response · Authors · 2025-12-03
> >
> > ### Strengths
> > We appreciated the general consensus among reviewers that the paper was particularly strong along the following dimensions:
> > - **Clear theoretically-founded conceptual insights** [nArH, bJDc, yHrP]:
> >   - "The theoretical analysis is rigorous, clearly attributing the diversity loss in RL-tuned LLMs to the mode-seeking behavior induced by the Reverse KL divergence."  [nArH]
> >   - "The paper formalizes the filtered target distribution, pc, and proves that maximizing the usual RLVR objective equals minimizing the reverse KL to a softened filter. This clarifies why RLVR becomes mode seeking and why diversity collapses in practice. The argument is simple but convincing." [bJDc]
> >   - "Clear conceptual contribution: The paper provides a unified perspective that clarifies how RLVR methods implicitly optimize Reverse KL to a filtered distribution, explaining diversity loss in a principled way." [yHrP]
> >   - "Theoretically grounded: The connection between RLVR and distributional matching (Lemmas 1-2) is well-established, and the use of α-divergences to interpolate between objectives is mathematically sound." [yHrP]
> > - **Simple unifying method** [m7ni,bJDc,nArH]
> >   - "A simple approach with meaningful peformance gain" [m7ni]
> >   - "The method unifies rejection sampling fine-tuning at one end and RLVR-style training at the other end, and allows smooth interpolation between mass covering and mode-seeking regimes. This aligns with prior distributional-matching insights and provides a knob practitioners can use." [bJDc]
> >   - "[The paper] addresses a meaningful research question, highlighting the significance of the accuracy–diversity trade-off in RL-aligned LLMs." [nArH]
> > - **Comprehensive experimental analysis** [m7ni,bJDc,yHrP]
> >   - "Extensive ablations" [m7ni]
> >   - "The paper also measures tactic and premise diversity and shows how those relate to pass@1 and pass@256, which is informative for understanding diversity effects beyond one score." [bJDc]
> >   - "Comprehensive experimental analysis: The diversity analysis using Simpson index and Shannon entropy, along with problem difficulty transitions, provides valuable insights into model behavior." [yHrP]
> > - **Presentation and clarity** [m7ni,nArH]
> >   - "Paper is well-written and easy to follow, with enough background explanation" [m7ni]
> >   - "The paper is well-structured..." [nArH]

---

> ### Author Response · Authors · 2025-12-03
>
> ### Weaknesses
> **Single model / single task experimental validation**  [m7ni, bJDc, nArH, yHrP]
>
> There was a general consensus that the paper would have benefited from showing that the proposed technique was effective in more than one model and dataset. We have addressed these concerns with **additional experiments** in Appendix G. First, we train a Qwen1.5B model on informal mathematical reasoning. There, we again see a diversity reduction effect, but smaller, for GRPO, which now has about the same coverage as the base model. Notably, Rw-Ulkly [4] also shows a small effect, being on par with RLOO [11]. $\alpha=0.999$, consistently with the previous experiment, performs similarly to these models. On the other hand, $\alpha=0.9$ achieves the highest pass@256, outperforming all other models, including Pass@k-Training. Lower values of alpha do not perform better on pass@256, which is puzzling for us. We conjecture that this comes from the fact that, this time, we did not pre-compute the partition function but rather computed it online on the basis of just 4 samples. We will confirm whether this is the case by adding an ablation experiment for the final version. Second, we have included additional experiments using the Kimina-Prover-Distill-1.7B model trained on Kimina-Prover-Promptset data. These are much more complex problems that require 8x longer sequence lengths and are more computationally intensive. For this reason, we have preliminary results only for GRPO, $\alpha$-DPG ($\alpha=0.5$) and the base model, observing as in previous cases that $\alpha=0.5$ achieves a higher coverage compared both to GRPO and the base model.
>
> **Missing baselines**  [bJDc, nArH, yHrP]
>
> Different reviewers suggested different additional baselines. bjDc suggested **rewarding the unlikely** [4] and **pass@k training** [6,7], which are two diversity-enhancing baselines. nArH suggested more RL-specific baselines such as **REINFORCE** and **PPO**. Reviewer yHrP suggested **DAPO**, **Dr. GRPO** and **GPG**. The suggestions from bjDc were somewhat puzzling to us because we were in fact evaluating the suggested baselines. In case the reviewer was arguing that we did not tune them well-enough, we argued that we used the same hyperparameters as the original papers and reproduced the same reported results and so, these were strong baselines. To reviewer nArH we expressed that REINFORCE was a weak baseline due to its high variance, and PPO is computationally complex due to requiring an additional critic network and here we focus on critic-free approaches which are also the state-of-the-art, and to reviewer yHrP we expressed that our GRPO baseline was in fact Dr. GRPO, and that we preferred to focus on stronger candidates than DAPO, which was a slight variation of GRPO.
>
> We still acknowledged the common feeling from multiple reviewers that more baselines could have been evaluated and **focused our efforts on 3 strong candidates**. For this we chose **GPG** [10], **RLOO** [11], and **ReMax** [12]. Of these, RLOO and ReMax had surprisingly good performance, achieving Pareto-optimal results, whereas GPG performed worse than GRPO. However, these results did not change our conclusions that $\alpha$-DPG produces models at the Pareto frontier, also achieving levels of coverage that are not reached by other methods.
>
> **Novelty** [m7ni, yHrP]
>
> Our paper revisits arguments from [2] in diagnosing the reasons for reduced diversity of KL-regularized RL, adopts the distributional framework of [1] to model problem-solving as constraints, and uses  $f$-DPG introduced by [3] to implement the proposed $\alpha$-DPG algorithm. Because the paper strongly builds on these prior studies, two reviewers mentioned a relative lack of novelty (even though m7ni did not weigh this very strongly, stating that the simplicity of the approach overshadows the “lack” of novelty and scoring the paper with 8). We argued:
>
> - **Incrementality is not an indicator of impact**: We concur that our work is incremental, yet many incremental innovations have been hugely impactful in the field (e.g. PPO could be seen as an incremental modification of TRPO, and GRPO as an incremental modification of PPO).
> - **There is still valuable novelty**: $\alpha$ divergences are not mainstream in ML and we are the first to bring them to the space of post-training LLMs.
> - **We strongly contribute to the conversation in the field with a viewpoint that has been completely neglected**: While the distributional techniques on which we build (and which make our work incremental) have existed for a few years and are relatively well-cited, there is **no comparison** to them in recent work. Yet, they offer a markedly different perspective from RL, which is the dominant paradigm. Thus, by putting the spotlight on distributional techniques, extending them with $\alpha$ divergences, and showing that they achieve state-of-the-art results, we believe we bring a strong contribution to the conversation in the field.

---

> ### Author Response · Authors · 2025-12-03
>
> [1] Khalifa, Muhammad, Hady Elsahar, and Marc Dymetman. “A Distributional Approach to Controlled Text Generation.” arXiv:2012.11635. Preprint, arXiv, May 6, 2021.
>
> [2] Korbak, Tomasz, Hady Elsahar, Germán Kruszewski, and Marc Dymetman. “On Reinforcement Learning and Distribution Matching for Fine-Tuning Language Models with No Catastrophic Forgetting.” arXiv:2206.00761. Preprint, arXiv, November 14, 2022.
>
> [3] Go, Dongyoung, Tomasz Korbak, Germán Kruszewski, Jos Rozen, Nahyeon Ryu, and Marc Dymetman. “Aligning Language Models with Preferences through f-Divergence Minimization.” arXiv:2302.08215. Preprint, arXiv, June 6, 2023.
>
> [4] He, Andre, Daniel Fried, and Sean Welleck. “Rewarding the Unlikely: Lifting GRPO Beyond Distribution Sharpening.” arXiv:2506.02355. Preprint, arXiv, June 20, 2025.
>
> [5] Yue, Yang, Zhiqi Chen, Rui Lu, et al. “Does Reinforcement Learning Really Incentivize Reasoning Capacity in LLMs Beyond the Base Model?” arXiv:2504.13837. Preprint, arXiv, May 16, 2025.
>
> [6] Chen, Zhipeng, Xiaobo Qin, Youbin Wu, et al. “Pass@k Training for Adaptively Balancing Exploration and Exploitation of Large Reasoning Models.” arXiv:2508.10751. Version 1. Preprint, arXiv, August 14, 2025.
>
> [7] Tang, Yunhao, Kunhao Zheng, Gabriel Synnaeve, and Rémi Munos. “Optimizing Language Models for Inference Time Objectives Using Reinforcement Learning.” arXiv:2503.19595. Preprint, arXiv, August 17, 2025.
>
> [8] Cichoki and Amari, Families of Alpha-, Beta- and Gamma-Divergences: Flexible and Robust Measures of Similarities, Entropy, 2010.
>
> [9] Wang, Haiming, Mert Unsal, Xiaohan Lin, et al. “Kimina-Prover Preview: Towards Large Formal Reasoning Models with Reinforcement Learning.” arXiv:2504.11354. Preprint, arXiv, April 15, 2025.
>
> [10] Chu, Xiangxiang, Hailang Huang, Xiao Zhang, Fei Wei, and Yong Wang. “GPG: A Simple and Strong Reinforcement Learning Baseline for Model Reasoning.” arXiv:2504.02546. Preprint, arXiv, May 1, 2025.
>
> [11] Ahmadian, Arash, Chris Cremer, Matthias Gallé, et al. “Back to Basics: Revisiting REINFORCE Style Optimization for Learning from Human Feedback in LLMs.” arXiv:2402.14740. Preprint, arXiv, February 26, 2024.
>
> [12] Li, Ziniu, Tian Xu, Yushun Zhang, et al. “ReMax: A Simple, Effective, and Efficient Reinforcement Learning Method for Aligning Large Language Models.” arXiv:2310.10505. Preprint, arXiv, May 16, 2024.

---

### Meta-Review · Area_Chair_xzo3 · 2026-01-07

**Summary:**

This paper presents a clear and theoretically grounded reframing of RL with verifiable rewards as distributional matching, and convincingly argues that reverse-KL optimization explains diversity collapse in reasoning-tuned LLMs. The proposed use of $\alpha$-divergences is conceptually clean, unifies several prior approaches, and empirically demonstrates a strong precision–coverage Pareto frontier, with particularly compelling gains in coverage on formal theorem proving. Reviewers broadly agreed on the soundness, clarity, and insight of the work, and the authors substantially strengthened the paper during rebuttal with added baselines, variance analysis, and complementary experiments. While the method itself is incremental relative to prior distributional matching and f-DPG work, the synthesis and empirical evidence make it a valuable contribution. That said, the core empirical observations are still driven primarily by a single main task and benchmark (Lean theorem proving), with additional results remaining limited in scope and preliminary. I recommend acceptance, but ask the authors to explicitly acknowledge in the paper that their strongest claims and observations are currently validated mainly on this task and may not yet generalize broadly beyond it.

**Reviewer Concerns:**

*Concerns largely addressed by the rebuttal*

1. Baseline completeness and strength: The authors added several strong RL baselines (GPG, RLOO, ReMax) and clarified equivalences (e.g., GRPO vs Dr. GRPO), showing that $\alpha$-DPG remains on or near the Pareto frontier across these comparisons.

2. Statistical significance and robustness: The rebuttal fixed evaluation bugs, added bootstrap variance estimates, statistical significance tests, and perplexity analyses, and explicitly discussed instability at low α with mitigation (reward clipping). Reviewers acknowledged these fixes as satisfactory.

3. Theoretical clarity of $\alpha$-interpolation: Additional formal analysis clarified how $\alpha$-divergences interpolate between mode-seeking and mass-covering regimes when supports differ, addressing requests for deeper justification beyond edge cases.

4. Missing details (e.g., partition function computation, relationship to Rényi/Cressie–Read divergences) were added, and the separation between prior work and novel contributions was clarified.

*Concerns partially addressed or still outstanding*

1. Breadth of empirical evaluation / generalization: While the authors added informal math (MATH -> Minerva) and preliminary Kimina-Prover results, the strongest evidence still centers on Lean theorem proving. Generalization to other tasks (e.g., code generation) and larger model families remains limited and is acknowledged as future work.

2. Novelty (incrementality): The rebuttal argued convincingly for conceptual and practical value despite incremental algorithmic novelty, but this concern is inherently subjective and remains a residual weakness for some reviewers.

3. Practical guidance for $\alpha$ selection/scheduling: The authors discussed possible strategies and stability considerations, but did not provide a definitive, task-agnostic procedure; this remains an open direction rather than a resolved issue.

**Reviewer Scores:**

Reviewer m7ni: 8 to 8
Reviewer bJDc: 6 to 6
Reviewer nArH: 4 to 5
Reviewer yHrP: 4 to 4

---

### Decision · Program_Chairs · 2026-01-26

Accept (Poster)